# The evolution of built-up areas in Ghana since 1975

**Marcel Fafchamps**[1]☺*, **Forhad Shilpi**[2]☺

**1** Stanford University, Stanford, CA, United States of America, **2** The World Bank, Washington, DC, United States of America

☺ These authors contributed equally to this work.
* fafchamp@stanford.edu

## Abstract

We use high resolution satellite data on the proportion of buildings in a 250x250 meter cell to study the evolution of human settlement in Ghana over a 40 year period. We find a strong increase in built-up area over time, mostly concentrated in the vicinity of roads, and also directly on the coast. We find strong evidence of agglomeration effects both in the static sense—buildup in one cell predicts buildup in a nearby cell—and in a dynamic sense—buildup in a cell predicts buildup in that cell later on and an increase in buildup in nearby cells. These effects are strongest over a 3 to 15 Km radius, which corresponds to a natural hinterland for a population without mechanized transportation. We find no evidence that human settlements are spaced more or less equally either over the landscape or along roads. This suggests that arable land is not yet fully utilized, allowing rural settlements to be separated by areas of un-farmed land. By fitting a transition matrix to the data, we predict a sharp increase in the proportion of the country that is densely built-up by the middle and the end of the century, but no increase in the proportion of partially built-up locations.

## Introduction

The world population has been growing rapidly. This is particularly true in Africa where the population has multiplied manifold since the beginning of the 20th century, and is predicted to reach 4.3 billion by the end of the 21st [1]. Vollset et al. make lower predictions overall, but still predict a large population increase in Africa during the first half of the century [2]. Increases in the African population have spurred urbanization in general, and the emergence of mega-cities of several million inhabitants in many—if not most—African countries [3, 4]. The rapid expansion of population created demand for housing, infrastructure and services, provision of which in turn will determine the future productivity and well-being of the population. African cities are crowded and disconnected with disproportionately higher living costs and lower economic density in terms of income [4]. Though housing conditions in much of Africa improved considerably between 2000 and 2015, more than half of Africa's urban population live in settlements with poor housing condition [5].

**Data Availability Statement:** Data are available from openICPSR: https://www.openicpsr.org/openicpsr/project/138201/version/V1/view.

**Funding:** The research was funded by the World Bank, of which one of the authors, Forhad Shilpi, is an employee. All views expressed in the paper are

those of authors and should not be attributed to the World Bank or its affiliates. The World Bank played an important role in helping us put the data together from various public sources. The World Bank had no role in the analysis, decision to publish, or preparation of the manuscript – other than the direct involvement of one of its employees, Forhad Shilpi, in the writing of the manuscript. This does not alter our adherence to PLOS ONE policies on sharing data and materials.

**Competing interests:** No author has any competing interest.

The expansion of urban land cover to accommodate a rising population has been fastest in Asia and Africa [6]. This raises concerns about irreversible changes in land cover leading to deforestation, loss of bio-diversity, proliferation of human settlements in disaster-prone areas, and other environmental damages [7–9]. How the rapid expansion of human settlements unfolds over geographical space is thus of paramount importance for development [10] and has likely deep consequences for inequality, poverty [11], climate mitigation, and urban planning [7]. While these structural changes create formidable challenges for policy makers [12], relatively little is known—beyond the above general observations—on the process by which urbanization is unfolding in the African continent overtime. Do partially buildup areas expand proportionally with an increase in buildup areas? Do newly buildup areas grow closer to already built-up areas or scatter over the landscape? Do rural towns and smaller cities emerge in regular intervals in relation to each other? These are the questions that we address in this paper.

Much attention has recently been devoted to the growth of cities in lower and middle income countries [12, 13] focusing mostly on the larger cities. A number of studies have examined the form and compactness of cities [14, 15] and their impacts on urban productivity [16–18], as well as the role of urban transit [19, 20] and land tenure regulation [21, 22] in shaping the urban space in developing countries. Several studies have used satellite and remote sensing data to document and forecast expansion of urban land cover at the global scale [6–9, 23] and for West Africa [24]. However, research on the evolution of lower density cities/towns has been limited despite its importance for policies regarding provision of infrastructure, housing and other services. Indeed, if partially built-up areas grow disproportionately over time—e.g., because of haphazard growth of urban areas or agricultural intensification—this creates different needs for public infrastructure investment and market development than if population growth concentrates in a few high density areas. It would also dictate a different spatial and size distribution of bank branches and retail outlets. We address this gap in the literature by examining the evolution of human settlements over the entire geographical space, focusing specifically on partially buildup areas.

We look into the question using detailed data on building density across space and time. We take advantage of the recently available Global Human Settlement Layer (GHSL) data from the Joint Centre of the European Commission, that uses machine learning tools to predict the presence of buildings from satellite imagery [25]. We focus on Ghana, a West African country that has experienced steady growth in population and GDP, without incurring any civil conflict since independence in 1957. Appendix C in S1 Appendix provides detailed background information about Ghana, its history, and its geography. At the heart of this study is a panel dataset of 3.8 million rectangle-shaped cells covering the entire territory of Ghana and observed in 1975, 1990, 2000, and 2014. For each of these cells, the GHSL dataset has an ML prediction of the proportion of that cell's area that is covered by buildings. We regard the existence of buildings as indicating the presence of people and economic activity—although we do not have direct evidence on either. Given that we have precise geocoordinates for each cell, we can combine this large dataset with information on roads, humidity, soil types, and proximity to borders and bodies of water. Appendix C in S1 Appendix provides additional information about the data.

With these data we investigate an issue understudied in LDCs: the spatial evolution of population density, proxied by buildup. From the literature we know that partial buildup—a special feature of our data—arises mainly in two settings: at the periphery of towns and cities; and in farmed areas. The urban periphery attracts a mix of peri-urban farming and urban commuters in search of cheaper housing [26, 27]. The result is a low density buildup in the vicinity of densely built urban centers. As towns grow, low density areas can grow around them, or the

city can crowd out its partially built hinterland. Regarding farmland, we expect population growth to lead to agricultural intensification [28] and to the emergence of a non-farm sector and rural market towns [29, 30]. We therefore expect rural towns to have a small densely-built center where non-farm production concentrates, surrounded by low density buildup made of a mix of farms and residences for non-farm producers. Given that rural towns are small, rural densification should show up as an increase in partially built-up areas. How fast low density areas grow relative to high density urban centers is an issue of interest to city planners.

As with the expansion of urban areas, the patterns of human settlements may also evolve in response to population growth leading to the emergence of new settlements over the geographical space. To study this, we focus on the spacing of human settlements, drawing from insights on the spacing of rural towns from the theoretical work by von Thunen [31], Christaller [32] and Isard [33]. Starting from von Thunen's model of rural market towns serving the surrounding agricultural hinterland, Christaller [32] and Isard [33] derive a network of rural towns where towns of varying size are spaced regularly. Salop offered a similar model on a road network [16]. The advantage of regular spacing of towns/human settlements is that it can simplify regional planning by allowing forecasting of future settlement patterns, thereby facilitating the positioning of infrastructure. We look for regularity in the spacing of human settlements, either on a plane, or along roads, at different points in time to detect changes in its pattern.

Our empirical analysis produces four key results. First, the average buildup density in Ghana increased from 0.4% in 1975 to 1.8% in 2014. The overall growth rate in built-up cell is about 1 percent per annum which is smaller than 1.3% growth rate in large cities between 2001–2018 reported by Sun et al. [23]. We find that at the very smallest level—i.e., partial buildup of a small geographical cell, Zipf's Law does not apply [34, 35]: the log of the buildup proportion of cells is far from forming a linear relationship with the log of the rank of that buildup proportion. This result suggests that growth of buildup at the micro level is not independent of its initial built-up level, providing suggestive evidence of possible persistence in buildup areas. More importantly, we find that average buildup density evolved in a way so as to leave the proportion of partially built-up cells constant. This suggests that urban growth in Ghana has not taken the form of disproportionate expansion of partially built cells. Rather, towns and cities crowded out their hinterland, probably because of commuting constraints. The evidence also suggests a large increase in small and medium urban settlements: urbanization happened not just in a few large cities. This is consistent with the findings from population growth as well: Ghana's urban population growth has been faster in small cities than its large ones [36].

Second, the evolution of built-up areas is highly correlated with roads. Roads attract a disproportionate share of the increase in build-up: only 14.24% of cells are located at most 1.5km from a road in 1976, yet they attract approximately 62.6% of buildings. Buildup forecasts based on fitted transition matrices confirms that proximity to roads is associated with a much larger increase in future buildup. Evidence shows that new roads locate near already builtup areas and subsequent construction locates near newly built roads. More research is needed on the causal effect of roads using quasi-natural experiments such as those used by [37, 38].

Third, we find that buildup in a cell predicts more buildup in cells located up to 3 to 5 km, indicating the presence of agglomeration externalities. For cells located near roads, there is still some evidence of more buildup up to 40–50 km away. For cells not near a road, the effect fades away faster, consistent with the idea that transport costs constrain the size of the town's hinterland. We find no evidence that densely built areas locate at regular intervals. One possible interpretation is that arable land has not yet been utilized enough for rural settlements to be crunched near each other.

Fourth, analysis based on the transition matrices shows that once a cell gets built up, it seldom reverts to an unbuilt state. If Ghana were to continue evolving as it did between 1975 and 2014, there would still be 91.9% unbuilt cells by 2055. But the share of the densely built-up areas would increase by 130%—admittedly from a very low base. Ghana will witness much growth in secondary towns and rural market hubs as well.

This paper contributes to the literature on urbanization and regional development [3, 6–9, 13, 23, 24]. Seto et al performed a meta-analysis of urbanization between 1970 and 2000 in developing countries and found the urbanization rate in Africa to be one of the fastest [3]. They find that urban land expansion rates are higher than or equal to urban population growth rates, suggesting that urban growth is becoming more expansive than compact during their study period. Similar findings are reported in [6–9, 24]. Evidence in Seto et al [3] shows that urban land expansion in Africa is driven largely by population growth and has a weak relationship with GDP growth. Sun et. al [23] confirms higher population growth in large cities in the low-income and lower-middle-income countries but find buildup area expansion to be more concentrated in the upper-middle-income countries. Henderson and Turner [13] also report that countries in Sub Saharan Africa is urbanizing early in the sense that they are urbanizing at levels of per capita income generally far lower than when developed countries urbanized. We contribute to this literature by showing that in Ghana—a country with relatively vibrant economy – urban buildup growth over 1975–2014 has been substantial but urbanization has not been associated with disproportionate increase in partially built-up areas. Several studies on Ghana find evidence of substantial urban sprawl around major cities in Ghana [39–41]. The growth of built-up areas attests to cities encroaching on their hinterlands in our analysis too. Our finding that the share of partially built-up areas remains constant overtime implies that the population in Ghana are congregating to cities and towns instead of spreading out into partially built-up areas.

The availability of GHSL data encouraged new research on urban areas at a much finer resolution. The fine grained data have been used more widely to define what constitutes urban areas (see, for instance, http://atlasofurbanexpansion.org/). Bellefon et al. [42], for instance, use buildup proportion as a way of identifying the edges of cities, arguing that partial buildup identifies sub-urban areas. Henderson and Turner combined GHSL data with household surveys data and find that urban living comes with benefits such as higher income and better access to services, and higher costs (crime, diseases. poor child outcomes) [13]. We adopt a philosophy similar to Bellefon et al. [42] in the sense that we regard building concentration as an indication of a dense human settlement in an urban setting. Our evidence also highlights the close association between human settlement, urbanization, and road infrastructure.

Recent literature has shown a substantial impact of transport infrastructure and technology on the geographical concentration of economic activities [37, 38, 43–45]. Much of this evidence is at a fairly aggregate level, e.g., a city or town. We show that a similar pattern exists for smaller settlements of a radius of a few hundred meters. Henderson et al. [46] show that, in countries that developed early, cities located in agricultural regions because of high transport costs, a pattern of agglomeration that survived after transport costs fell. In a similar vein, Jedwab [47] documents how, in Ghana and Côte d'Ivoire, cocoa production spawned many small towns and how these towns survived the gradual westward shift of cocoa plantations that took place over the last century. Our results confirm that buildup is highly persistent: once a building appears in a cell, it is very rare for this cell not to also have a building in subsequent years, even though we cannot say if it is the same building. Moreover, places that are already settled tend to attract additional buildings, meaning that they become a focal point for population growth. The same can be said of roads, in the sense that proximity to a road in 1976 or 1986 is a strong predictor of human settlement in 2014, even when that road has since disappeared.

An intellectually elegant literature, spearheaded by von Thunen in [31], argues that human economic activity self-organizes over space into equally spaced settlements. Similar ideas have been applied to the distance from which workers are willing to commute, and to study the geographical coverage of shopping malls and grocery stores. Some evidence has been provided that towns and cities are more or less equally spaced from each other—for African towns along railway lines [29, 30]; and for Chinese cities along newly built freeways [38]. In both of these cases, however, planning by a government or colonial authority seems to have been influential. We find no such evidence in Ghana over the study period, something we interpret within the theory as implying that the full utilization of arable land has not been reached yet.

We start in Section 2 on methods and materials by introducing our conceptual framework, followed by the description of the data construction and the empirical methodology. Section 3 presents and discusses the empirical results. Section 4 concludes.

## Materials and methods

### Conceptual framework

The literature on regional and urban economics is, in one way or another, about where people and economic activity locate and how this changes over time. In this paper we revisit this issue using a newly available, highly dis-aggregated dataset on buildup in a West African country. While this dataset has considerable breadth in terms of coverage and geographical detail, it also lacks many variables that have received attention in the literature—such as population, jobs, sectors of activity, production, and infrastructure. This forces us to adopt a reduced-form approach that emphasizes breadth instead of depth.

To begin, we regard the presence of buildings in a cell as indicating the presence of population and, hence, of economic activity nearby. The denser buildup is in a cell, the more people live in that cell, deriving a livelihood from work in the vicinity. This starting point is unproblematic given the level of generality of our analysis. Our data contain lots of cells that are only partially built. Hence, in order to make full use of our data, we need to put together a conceptual framework on partially built-up cells that is inspired from the literature while remaining sufficiently sparse to match the limited depth of our data. This conceptual framework also must include all cells – not just metropolitan areas, which often are the sole focus of spatially dis-aggregated empirical analysis [17, 18, 21, 48–50].

The lowest level of building density—e.g., a lone building surrounded by unbuilt cells—is probably a farm. Ghana is a large agricultural country with a relatively undifferentiated geography, but a strong rainfall gradient running from North West (semi-arid) to South (humid). More details are provided in Appendix C in S1 Appendix. Since humid areas can support a higher population density than dry ones, we expect a higher frequency of farms in the South. More generally, agricultural potential varies with agro-ecological features and so does agricultural population density. What is less clear is whether, in denser areas, farms cluster in villages—in which case buildup is concentrated in a few cells – or whether they disperse across the land—in which case there are many cells with a single homestead on them. Which pattern best describes Ghana's rural population is not entirely clear—some factors militate in favor of clustering (e.g., for defense purposes during the slave trade); others in favor of dispersion (e.g., long fallows). Data on partially built-up cells provides an easy way of characterizing this distribution.

Population growth has also fostered a process of agricultural expansion whereby land previously un-farmed gets newly settled. This process is common to much of sub-Saharan Africa, and was still underway during our study period. We therefore expect that, over time, new farms spring up in previously empty cells. What is unclear is whether these new farms form

new clusters, or whether they spread out more or less evenly over the landscape as time passes. The answer to this question is of obvious interest to policy markets interested in rural infrastructure and agricultural marketing: it is a lot easier to provide agricultural and public services to a population that is relatively concentrated. To cast light on this question, we examine how the distribution of partially built-up cells evolves over time: do we observe a gradual dispersion of buildings across the land, or a gradual concentration. We also investigate what is the long-term trend in terms of building (and thus population) dispersion across the country. This is done by constructing the transition matrix of built-up areas and using it to predict buildup distributions by the middle and the end of this century. This research question is related to well-studied processes in developed countries, such as rural flight and and its relative converse, urban sprawl [51].

A closely related issue is the location of rural market towns. As rural population increases, a demand for specialized services arises (e.g., retail shops, hair salon, bicycle repair) that favors the emergence of rural towns [28]. Rural settlements can also be jump-started by investments in agricultural infrastructure and production [52] or mining [53]. Sub-Saharan Africa—as in other LDCs—has experienced a rapid rise in the growth of rural non-farm enterprises, often in the form of small shops and artisans [41]. This process of rural small-scale urbanization—and the accompanying distribution of human settlement across space—was already the focus of von Thunen's work [31], and has since attracted the attention of numerous others [26, 27, 32, 33]. The common thread running through this literature is the concept of rural hinterland: each market town serves a surrounding area, the diameter of which depends on transportation costs. For instance, if farmers will only walk two hours to sell their crops and purchase retail goods, the diameter of the hinterland of a market town is two hours walking time. In this case, we expect the landscape to be characterized by small market towns surrounded by farms scattered over a radius of a few kilometers at most. We show in Fig A1 in S1 Appendix what this pattern of settlement may look like, together with a spatial covariogram that shows, for each pairwise distance between cells, the proportion of pairs made of two built-up cells. We note that, at close distances, there is an excess of built-up pairs. This captures the strength of agglomeration forces: buildings are located in and around small local hubs. At longer distances, there is no systematic pattern because settlements are located a random distances from each other.

If population density is low, rural hubs can locate at will—e.g., in locations that have good access to arable land and drinking water. When density gets high enough, however, the hinterland of these rural hubs start to overlap. This generates competition between them at the margin, e.g., to attract farmers selling their crops and buying consumer items. Provided that rural consumers enjoy a diverse offer of goods and services in a single location, rural market towns – like shopping malls—cannot locate immediately next to each other. This results in a regular spacing of rural towns, either on a plane (in which case hinterlands form a honeycomb shape with market towns at the center—[33]) or along a road [16, 38] or railway line [45]. The hinterland concept is illustrated in Fig A2 in S1 Appendix where we show, on the left, a simulated example with an equal-spacing distribution of rural hubs. There is some limited evidence that rural towns are fairly equally spaced in Africa [30] but this theory has never been tested at the level of an entire country. Regular spacing of small market towns on a plane would show up in a spatial covariogram as a spike at regular intervals, as shown in the right-hand graph of Fig A2 in S1 Appendix. Regular spacing along roads would show the same signature when limiting the attention to cells along roads. We test these ideas in the empirical part of the paper. We do not expect this regular spacing to hold when rural population density is too low.

In our data, small rural market towns may occupy one or two cells at most. What of larger settlements, what distribution of buildup cells can we expect to observe? While there is a lot of

literature on the growth of large cities (see [3] for a review), less attention has been devoted to the growth of urban settlements of small and middle sizes, especially on the African continent. In particular, little is known regarding the expansion of partially built-up cells during rapid urbanization: do towns and cities grow as densely packed entities; or do they spawn a large hinterland of peri-urban agriculture [54] or urban commuters [55]? To articulate these ideas formally, we develop a toy model of cities as concentric circles: at the middle of the city, all cells are fully built according to our data definition (i.e., cells contains buildings, roads, sidewalks, parking lots, and the like). This center is then surrounded by a sequence of donuts of different sizes characterized by a decreasing density of buildup (see Fig A3 in S1 Appendix, top left). We think of falling buildup density as resulting from a varying mix of sub-urban residences for commuters—which can be densely packed—and those of farmers who provide urban consumers with vegetables, fruits, and dairy products and need land to farm. As distance to the center grows, fewer commuters move in and more land is left for farming. This is obviously a simplification, but it is sufficient for our purpose.

How do the donuts change as the town grows? Regarding farmers, we can imagine that demand for peri-urban crops increases more or less proportionally with population and thus with the size of the town center. Similarly, we can expect the number of town workers to increase more or less proportionally with town size—and thus the number of aspiring suburban commuters. This implies that if the town center – made of fully built-up cells—expands by a factor of $k$, so does the area covered by the partially built-up donuts that surrounds it. To see this, let $C$ denote the fully built-up area at the town center. We have $C = \pi r^2$ where $r$ is the radius of the center. Let $R$ denote the outer radius of a partially built donut that surrounds the town center. In the discussion here we imagine a single donut, but the same reasoning applies to multiple donuts of varying density.

Imagine that the area of the center grows by a factor $k$. By how much must $R$ increase for the area of the donut to also increase by $k$? Since the area of the donut is $\pi R^2 - \pi r^2$, the answer is $k$. We want the ratio of the donut area to the center area to remain constant, i.e., we want $\frac{\pi R^2 - \pi r^2}{\pi r^2}$ = constant when $r$ is set to $kr$. It is immediately clear that this requires that $R$ be multiplied by the same $k$. This means that, in this world, the proportion of partially built-up land around towns and cities increases at the same rate as the fully built-up land. This prediction is testable with the data we have. If we combine this prediction with the rise of partially built-up cells due to agriculture and small market towns that we discussed earlier, we expect the fraction of total land that is partially built-up across the country to increase at the same rate or faster than the fraction of fully built-up cells. One specific version of this prediction is Zipf's Law applied to the size of human settlements. The size distribution of cities is said to follow Zipf's Law if we observe a linear relationship between the log of city size and the log of the rank of a city in terms of size [34]. Applied to buildup in cells, Zipf's Law would imply a linear relationship between the log of buildup proportion and the log of the rank of the cell in terms of buildup. If Zipf's Law were to hold for buildup, it would also imply that the frequency distribution of buildup across cells would remain constant as urbanization takes place—which in our toy model could only arise if city growth implies proportional growth in the size of the hinterland where partially built-up cells are located. The reverse is, however, not true: the frequency distribution of buildup could be time-invariant but not follow a power law. We test this hypothesis as well.

So far we have assumed that the area traveled by commuters does not matter. As the town grows, however, suburbs move further out, and sub-urban commuters must travel a longer distance. Presumably, as $R$ keeps increasing, the length and duration of the commute increase in roughly the same proportion (if not more, e.g., due to increased congestion in the center).

Without investment in rapid transit infrastructure for commuters, this is likely to reduce the appeal of suburban living, convincing some would-be commuters to shift to high density residential areas closer to the town center. In this case, the radius of the outer donut grows less and the radius of the center grows more for a given population increase. In the extreme case where $R$ represents an absolute limit on commutable distance, urban growth is characterized by an increase in $r$ while $R$ remains unchanged. The town grows up to the point at which $r = R$ and the entire area is fully built-up, at which point it stops: the city has reached the limit set by the available transport infrastructure and technology [19, 26, 27]. A similar prediction holds even if some rapid transit infrastructure is introduced, e.g., in large cities, but it is insufficient to allow $R$ to increase as the same speed as $r$: the city still densifies, even if at a slower rate. These cases are illustrated in Fig A3 in S1 Appendix. Here the prediction is that the *ratio* of partially built-up urban cells over fully built-up cells falls as the proportion of fully built-up cells grows. This is because cities densify as they grow (i.e., the ratio of $\frac{R}{r}$ falls as $r$ increases). This is illustrated in Fig A4 in S1 Appendix, which shows the simulated frequency distribution of partially and fully built-up cells when a town grows as shown in Fig A3 in S1 Appendix. Initially there is a large fraction of partially built-up cells, especially low buildup density cells. As the town grows, the proportion of these cells falls while the proportion of fully built-up cells rises. But the proportion of built-up cells of intermediate density does not change much. In contrast, if a city's hinterland expands proportionally to the city center, the frequency distribution of partially and fully built-up cells remains unchanged—which implies that the proportion of sparsely built-up cells rises at par with fully built-up cells.

In the paper, we look at the long-term evolution of the frequency distribution of partially and fully built-up cell to throw light on whether towns and cities crowd out their hinterland as they grow, or whether their hinterland grows with them. We nonetheless recognize that a fall in the ratio of sparsely built-up cells to fully built-up cells due to crowding-out may be partially offset by growth in the number of rural market towns. Here too, however, the same logic applies for individual towns if commuting time is a constraint: as small settlements grow, they absorb their hinterland, and the fraction of partially built-up land falls there as well. Only the creation of a sufficiently large number of *new* low-density villages and rural market towns can compensate this evolution. Whether this took place in Ghana over our study period—and is likely to continue over the foreseeable future—is thus an empirical question.

The visual representation of the concept of hinterland used so far is spherical, which essentially assumes that travel time is the same in all directions. This may be a reasonable assumption for travel by foot or bicycle on the relatively flat rural terrain that characterizes much of Ghana. It does, however, fail to recognize the role played by roads in facilitating transport and commuting. Conceptually, roads can be thought of as reshaping space to make certain locations closer to each other (in terms of travel time or cost) than they were without it. We mentioned this idea earlier when discussing workers commuting from sub-urban areas: roads—and transit systems more generally – help shape cities [19]. Roads can also shape the locations of rural market towns and even farmers themselves, e.g., because they allow inter-city trade [56]. We take this into account in our empirical analysis by examining the long-term effect that past roads may have on the location of human settlements, while simultaneously examining agglomeration effects.

## Data

Our main variable of interest $y_{it}$ measures the proportion of a cell $i$ in year $t$ that is covered by buildings, as inferred from a high-definition satellite image of Ghana. This variable comes from the Global Human Settlement Layer (GHSL) data that have been put together by the

Joint Centre of the European Commission. A Mollweide projection was used to divide the earth's surface into square cells of approximately 250 by 250 meters, which is rather small for an analysis of built-up area. The proportion of buildings on each cell was predicted using a machine learning algorithm trained on observations of actual buildings made on the ground. Pesaresi et al. [25] describes the machine learning algorithm in detail. This algorithm predicts buildup in squares of 38x38 meters. These predictions are then aggregated using GIS into 250x250m cells, thereby generating an accurate prediction of the proportion of the cell that has buildings on it. Using code written in R, we overlay the GHSL data on a GIS map of Ghana and extract the cells that lie within the country's borders. A little over 3.8 million cells are needed to cover the entire area of Ghana. Additional information about the data is provided in Appendix D in S1 Appendix.

The GHSL dataset provides information on built-up cells for four separate years: 1975, 1990, 2000, and 2014. Cells for each of these years have been carefully aligned using detailed GPS coordinates so as to form a panel dataset for $y_{it}$. Aligning satellite photographs with each other and over time was achieved projecting each photograph using the same Mollweide projection. Variable $y_{it}$ takes values between 0 and 1, with 0 meaning no building detected in the cell, and 1 meaning the cell is entirely occupied by buildings—i.e., as in a city center. More precisely, $y_{it} = 1$ when all its area is covered by impervious surfaces, such as roofs, sidewalks, streets, and courtyards. As shown in Table 1, most cells (98.5%) in 1975 are predicted to be unbuilt, which is hardly surprising given that Ghana at the time only had a national population density of 43.9 per square km. Built-up area grows over time as population increases to 119.4 per sqkm in 2014 and Ghana becomes more urbanized (urban population share increasing from 30% in 1975 to 53% in 2014). In 2014, 93.8% of cells were unbuilt.

For each cell $i$ in year $t$ we construct two measures of buildup in the cells surrounding $i$. The first of these measures is denoted as $y_{it}^{n_1}$ and is the average built-up area in the cells immediately adjacent to $i$. Since the data is divided into a square grid, each cell on that grid in principle has 8 immediate neighbors, which constitute the set $n_1$. We thus have:

$$y_{it}^{n_1} \equiv \frac{1}{n_1} \sum_{j \in n_1} y_{jt} \tag{1}$$

where by an abuse of notation $n_1$ represents both the set of immediate neighbors and their number. Together with $y_{it}$, this variable captures the proportion of built-up area in an approximate cell 675 meters wide by 825 meters high (approximately 2025 by 2475 feet). Cells located on the border or along the sea or large body of water do not have a full complement of 8 adjacent cells located wholly or partly within Ghana. For these cells, the immediate neighborhood is limited to those cells included in the dataset and the average is adjusted accordingly. We also construct a variable $y_{it}^{n_2}$ that captures the average buildup in cells immediately adjacent to $y_{it}^{n_1}$.

**Table 1. Built-up area.**

| Year | Proportion of built-up area | | | Share of cells | Share of non-empty cells |
|------|------|------|------|------|------|
| | All | N1 cells | N2 cells | that are empty | that are fully built-up |
| 1975 | 0.43% | 0.43% | 0.42% | 98.50% | 1.96% |
| 1990 | 0.80% | 0.80% | 0.80% | 97.48% | 6.61% |
| 2000 | 1.19% | 1.19% | 1.19% | 96.52% | 8.41% |
| 2014 | 1.85% | 1.85% | 1.84% | 93.78% | 6.70% |

Notes: Total number of observations is 3,853,273. N1 cells are the 8 rectangles surrounding the cell. N2 cells are the 16 rectangles surrounding N1 cells. The number of N1 cells is 8 for all but 27,241 cells. The number of N2 cells is 16 for all but 54,324 cells.

There are in principle 16 cells in set $n_2$, except in the vicinity of borders and large bodies of water where that number gets truncated. The formula for $y_{it}^{n_2}$ is that same as Eq (1) but with $n_1$ replaced by $n_2$. Together, $y_{it}$, $y_{it}^{n_1}$ and $y_{it}^{n_2}$ cover an area 1125 meters wide and 1375 meters wide, roughly 1.55 square kilometers. These are generated by using spatial lag functions in R.

We combine data on built-up area with information on agro-ecological zones to capture the wide variety of landscapes and agricultural potential that characterize Ghana. The data for agro-ecological zones are downloaded from the GAEZ-Global Agro-Ecological Zones data portal that was developed by the Land and Water Division of the Natural Resources Management and Environment Department of FAO. The resolution for GAEZ dataset is 10 km. We overlay the 250 by 250 meter grid of the GHSL dataset over the GAEZ dataset to extract soil types for each cell. We divide this information into two variables, which do not vary over time. The first of these variables characterizes the humidity of a particular region, i.e., whether moist/dry, semi-humid, or humid. In Ghana as in the rest of West Africa, the rains follows a monsoon pattern and rainfall generally decreases along a North-South gradient, with the North being drier and the South being more humid. There are exceptions to this general pattern, however—the immediate coastal area, for instance, tends to be drier than inland. In the data, humid soils are associated with the entire valley of the Volta river and with the forested region North of Takoradi.

The second agro-ecological variable we construct captures soil quality – going from poor to moderately good to good. There are two additional categories: hydromorphic soil, which is characterized by excess moisture which tends to suppress aerobic factors in soil-building; and soils unsuitable for buildings or agriculture, such as steep soils or soils covered by standing water at least part of the year—these soils require special investments (e.g., terracing, drainage) in order to become usable. Given that Ghana is relatively flat with no mountains and a relatively homogenous physical landscape, we abstract from altitude: whatever variation in altitude there may be, it translates into variation in rainfall and soil quality and is already captured.

To this data we add longitude and latitude information of cells from the GHSL dataset expressed in decimal degrees, which we normalize to have mean zero and unit variance. A higher value of longitude means more to the West; a higher value of latitude means more to the North. Population density in Ghana is in general higher in the humid South than in the drier North, largely due to the higher agricultural potential and better soils in the South. But there is a lot of local variation, something we aim to capture with our set of agro-ecological and geographical variables.

We also have information about the presence on a cell of a land border, sea border, or large body of water from the GIS shape files of Ghana. We drop from the $y_{it}$ database of all the cells outside Ghana, as well as those that are entirely covered by water, since they cannot be built on. This includes lake Volta, a large artificial lake that covers 8,502 square kilometers within Ghana. Some cells are partially outside Ghana, or partially covered in water or by the sea, something we capture with dummies. We also construct a dummy equal to one if there is a large body of water on any of the eight adjacent cells $n_1$, and another one for the next 16 adjacent cells $n_2$. This is done to investigate whether proximity to a lake or lagoon attracts human settlement, e.g., if the lake is use to fish or transport goods. The same thing is done for proximity to the sea and to a land border, for similar reasons.

The last piece of information that we integrate into our analysis is road data for 1976 and 1986, at the beginning of our period of analysis. We digitized road maps from the Ghana population censuses. These maps are then used to extract information on the presence and/or absence of roads in GHSL cells. These roads are classified into four categories, from 1 to 4. There is also a class 5 which is labeled 'unknown road'. Out of an abundance of caution, we

ignored these roads in our analysis. Class 1 roads are the highest quality, and class 4 roads the poorest. Typically, a class 1 road is paved and hence usable all year round. In contrast, a class 4 is a dirt track and many of these roads are seasonal. Class 2 and 3 occupy an intermediate position. Road availability in 1976 is of particular interest for our purpose because these roads pre-date our period of analysis and thus cannot be construed as caused by buildings constructed after 1976. The same can be said of 1986 roads regarding new buildings constructed afterwards. Using the digitized maps, for each cell we construct a dummy $r_{kit}$ equal to 1 if there is a road of class $k$ in cell $i$ in year $t = 1976$ or $t = 1986$ and 0 otherwise. This is done by overlaying road maps on the GHSL cells defined earlier. Next we calculate the proportion of immediately adjacent cells that contain a road of each particular class, which we denote $r_{kit}^{n_1}$. Mirroring what we did for built-up area, we also calculate the proportion of roads in the $n_2$ neighborhood of $i$, which we denote $r_{kit}^{n_2}$.

## Empirical methods

Three types of analyses, described below, are used to shed lights on the evolution of buildup areas in Ghana.

**Forecasting buildup.** We start by estimating the frequency distribution of built-up cells over four years: 1975, 1990, 2000, and 2014. Nonparametric plots of the rank of partially built-up cells against the proportion of built-up area are used to check whether our data conform to Zipf's law.

To examine the long term tendency of buildup, we construct the transition matrix $M$ of $y_{it}$ between 1975 and 2014. Formally, let $f(y_{i,t=1975})$ be the frequency distribution of built-up cells in 1975, discretized into $N$ intervals; and let $f(y_{i,t=2014})$ be the same thing for 2014. The transition matrix $M$ is defined as the $N \times N$ matrix that satisfies:

$$f(y_{i,t=2014}) = Mf(y_{i,t=1975})$$

Row $d$ of $M$ provides the frequency distribution of $y_{i,t=2014}$ for all the cells that belonged in interval $d$ in 1975. For our analysis, $f(y_{i,t=1975})$ is divided into 11 groups: the first group includes all the unbuilt cells; the other ten are the deciles of the distribution of partially and fully built-up cells. Matrix $M$ can be constructed directly from the data.

Taking the full distribution of $y_{it}$ in 2014 as starting point, we then use $M$ to forecast the future distribution of built-up area in Ghana in 2053 and 2092 as:

$$\hat{f}(y_{i,2053}) = Mf(y_{i,2014}) \text{ and } \hat{f}(y_{i,t=2092}) = M^2 f(y_{i,t=2014})$$

Since these forecasts assume that the matrix $M$ of conditional transition probabilities remains unchanged over the forecasting period, they should thus be seen as a based on past trends in buildup between 1975 and 2014. We also calculate the ergodic distribution corresponding to transition matrix $M$ by solving equation:

$$\hat{f}(y_{i,ergodic}) = M\hat{f}(y_{i,ergodic})$$

Since $M$ is a stochastic matrix, its first eigenvalue is 1. It follows that $\hat{f}(y_{i,ergodic})$ is the first eigenvector of $M$. Distribution $\hat{f}(y_{i,ergodic})$ is not intended to be a forecast. Rather, it represents the long-term tendency in the distribution of built-up cells that is implied by $M$ and only serves to provide an easy visual representation of future buildup if past trends were to perdure indefinitely.

**Estimating agglomeration forces.** Next, we examine the extent of correlation between built-up levels $y_{it}$, $y_{it}^{n_1}$ and $y_{it}^{n_2}$ to get a sense of agglomeration forces in buildup. To this effect,

we estimate an OLS regression of the form:

$$y_{it} = \beta_0 + \beta_1 y_{it}^{n_1} + \beta_2 y_{it}^{n_2} + u_{it} \tag{2}$$

It goes without saying that coefficients $\beta_1$ and $\beta_2$ should not be given a causal interpretation; they simply document the spatial correlation patterns in buildup.

We then investigate whether cells in a built-up neighborhood subsequently experience more buildup. We start by estimating an OLS regression of the form:

$$y_{it} = \beta_0 + \beta_1 y_{it-s} + \beta_2 y_{it-s}^{n_1} + \beta_3 y_{it-s}^{n_2} + u_{it} \tag{3}$$

where $s$ is a time lag. We also estimate a non-parametric version of regression (3) in which the three regressors are replaced by dummies for each of the deciles of the relevant regressor. A vector $X_i$ of time-invariant cell characteristics is included as controls. These include soil quality and soil humidity categories; latitude, longitude, their cross-product; and dummies for the presence of a body of water, a land border, or a sea border. The estimated regression has the form:

$$y_{it} = \beta_0 + \sum_{k=1}^{10} \beta_1^k y_{it-s}^k + \sum_{k=1}^{10} \beta_2^k y_{it-s}^{kn_1} + \sum_{k=1}^{10} \beta_3^k y_{it-s}^{kn_2} + \theta X_i + u_{it} \tag{4}$$

where $y_{it-1}^k$ is a dummy equal to 1 for the $k$'th decile of $y_{it-s}$, and $y_{it-s}^{kn_1}$ and $y_{it-s}^{kn_2}$ are the same set decile dummies for $y_{it}^{n_1}$ and $y_{it}^{n_2}$. We only estimate regression (4) for the shortest lag $s$, i.e., using 1975 for 1990, 1990 for 2000, and 2000 for 2014. Eq (4) provides finer information on the strength of agglomeration effects.

We then expand the model to include dummies for different road categories in order to investigate the predictive role of past road construction on subsequent buildup. The estimated regression takes the form:

$$y_{it} = \beta_0 + \sum_{k=1}^{10} \beta_1^k y_{it-s}^k + \sum_{k=1}^{10} \beta_2^k y_{it-s}^{kn_1} + \sum_{k=1}^{10} \beta_3^k y_{it-s}^{kn_2} + \gamma_1 r_{it-s} + \gamma_2 r_{it-s}^{n_1} + \gamma_3 r_{it-s}^{n_2} + \theta X_i + u_{it} \tag{5}$$

where: $r_{it-s}$ is a vector of road availability dummies, one for each road type, at time $t - s$ in cell $i$; $r_{it-s}^{n_1}$ is a vector of average availability of different types of roads in neighborhood $n_1$, and $r_{it-s}^{n_2}$ is the same for neighborhood $n_2$. The coefficient $\gamma_1$ gives the predicted percentage points increase in buildup if there was a road in the cell at time $t - s$. Coefficients $\gamma_2$ and $\gamma_3$ give the predicted percentage points increases in buildup as neighborhoods $n1$ and $n2$, respectively, go from having no cell with a road at time $t - s$ to having roads in each cell.

**Spatial covariogram.** The regression analysis above examines agglomeration effects at fairly close range—i.e., up to 1.5km at most given that each cell is 250m × 250m. To explore whether there is regularity in terms of locations of clusters of built-up cells over a broader range of distances, we estimate a spatial covariogram of cell buildup. The spatial covariogram of a spatial variable $z$ plots the average correlation between observations of $z$ for various distances between them. This is in principle achieved by calculating the distance between all possible pairs of observations and then by computing, for each relevant distance interval, the sample correlations between $y$ pairs located within that distance interval from each other. The concept is similar to autocovariogram in time series analysis, except that it is calculated over distances which, in data like ours, are not limited to integer values.

To do this effectively, we have to overcome the challenge of dimensionality. With nearly 4 million cells in our data, calculating all pairwise distances would generate around 16 trillion observations—a number that far exceeds the computing capacity at our disposal. To

implement our plan, we therefore need to resort to a few tricks. The first trick is to focus on the spatial covariogram and simplify the calculation of covariance by focusing on a dichotomous variable $z_{it} \equiv Pr(y_{it} > 0)$. This means that the sample equivalent of the spatial covariance $E[z_{it} z_{jt}]$ at distance $d$ is simply the proportion of cell pairs $ij$ at distance $d$ that are both 1. Our objective is thus to plot a sample estimate of $E[z_{it} z_{jt}]$ at various distances $d$ for Ghana. To this effect, we need to construct a representative sample of cell pairs and calculate the proportion of pairs that are both built-up among all sampled pairs.

To achieve this, we divide Ghana into a number of tiles small enough that we are able to construct all the cell pairs in that tile. This is done by creating a grid of 856 equal size tiles that covers all of Ghana. In practice, this is achieved as follows. We first identify the extrema of latitude and longitude in Ghana. We then divide the North-South difference into 33 equal intervals. We do the same for the East-West difference. This creates 1,089 blocks, 856 of which cover at least part of Ghana; the rest fall in neighboring countries and are ignored here. We also drop a handful of tiles with a number of cells too small to compute a covariogram. This implies that we are not using all the data at our disposal: pairs of observations that belong to different tiles are not used to construct the covariogram. Since tile boundaries are chosen at regular intervals based on latitude and longitude, they are orthogonal to the phenomenon we study. Hence the resulting sample of cell pairs should be representative. Each of these tiles is an $18 \times 18$ km square (approximately 26 km in diagonal). It comprises around 5,200 cells and approximately 13.5 million unique cell pairs. We then divide cell pairs into approximately 260 distance bins of 100 meters each. The number of cell pairs in each bin varies with the shape of the tile and the presence of borders and water bodies. We then count, for each distance bin $b$, the total number of pairs $N_b = \Sigma_{ij \in b} C_{ijt}$ in that bin—where $C_{ijt} = 1$ for cell pair $ij$ and the summation is taken over all the cell pairs in bin $b$. We also count, for each year $t$ and each bin $b$ in each tile $m$ the number of pairs with two built-up cells, that is, two cells $i$ and $j$ in $b$ for which $z_{it} = 1$ and $z_{jt\,=\,1}$. This enables us to calculate the share $S_{btm}$ of cell pairs in tile $m$ and in distance bin $b$ that are both built up in year $t$ as:

$$S_{btm} = \frac{\sum_{ij \in b,m} z_{jt} z_{jt}}{N_{bm}}$$

Since this is done separately for each tile $m$, it means that the values of $S_{btm}$ can be computed separately tile by tile before assembling them together to cover all of Ghana.

Because country borders and bodies of water are irregular and the length of one degree of longitude varies a bit from North to South Ghana, the size and shape of each tile varies somewhat and so does the number of cell pairs in each bin of each tile. This means that to reconstruct the country-level covariogram, we need to weight each $S_{btm}$ by the number of cell pairs that it represents. To achieve this, we simply regress $S_{btm}$ for a given year on distance bin dummies:

$$S_{btm} = \sum_b \alpha_b D_{bm} + u_{btm}$$

and weight each observation $S_{btm}$ by the number of observations $N_{bm}$ in bin $b$ of tile $m$. We then plot, for each year $t$ the predicted values $\hat{S}_b$ (they only vary by distance bin $b$) and the prediction standard errors obtained from the above regression. This process yields a spatial covariogram averaged across all tiles and weighted by the number of cell pairs in each bin of that tile. The maximum distance within a tile is about 26 km. We only show confidence intervals for 1975 and 2014, to keep the graph uncluttered. As is immediately apparent from the graph, confidence intervals are extremely tight.

To investigate regularity in the placement of built-up areas at longer distances, we recalculate the spatial covariogram for distances up to 75 km. To do this, we divide Ghana into 86 tiles of roughly $60 \times 60$ km, with a diagonal distance of over 76km. Because the proportion of built-up cells is low, we are still able to form all built-up cell pairs and to compute $\sum_{ij \in b,m} z_{jt} z_{jt}$ for all but the 3 most densely built-up tiles, which contain little information of interest for our purpose. We cannot compute $N_{bm}$ directly, however, due to computer power requirements. We therefore must proceed differently. The total number $T_m$ of unique cell pairs in each tile is immediately obtainable: it is $T_m = N_m \times (N_m - 1)/2$, where $N_m$ is the number of cells in tile $m$. What we need is an estimate of how these $N_m \times (N_m - 1)/2$ are divided across distance tiles. Indeed, because tiles vary in shape, the distribution of pairs across distance bins varies across tiles. To construct an estimate of the frequency distribution of cell pairs into distance bins for each tile, we select in each tile a large but manageable random sample of cells. We then form all cell pairs among them and calculate the number of pairs in each distance bin $R_{bm}$. Let $R_m = \sum_b R_{bm}$ be the total number of cell pairs in the random sample for tile $m$. Since we did not use all cells in the tile, $R_m < T_m$. The frequency distribution of distance bins in each tile can nonetheless be approximated as:

$$N_{bm} \simeq R_{bm} \times \frac{T_m}{R_m}$$

where $T_m/R_m$ is used to rescale the distribution of weights to match the full size of the tile.

## Results and discussion

### Descriptive statistics

We start by showing in Tables 1 to 3 key descriptive statistics for our variables of interest. From Table 1 we see that proportion of empty cells ($y_{it} = 0$) fell 98.5% to 93.8% between 1975 and 2014. Averages are essentially identical for $y_{it}$, $y_{it}^{n_1}$ and $y_{it}^{n_2}$, which is hardly surprising since they average essentially the same cells. The proportion of fully built-up cells ($y_{it} = 1$) among built-up cells increased a lot between 1975 and 1990—from 2% to 6.6%, but does not show any trend after that. We revisit this point more in detail below. We check whether Zipf's Law applies to built-up area at this micro level. To do this, we sort partially built-up cells by the proportion of cell area that is built up—from largest to lowest. We then plot the log of the rank to the log of the built-up area. Here cells with a built-up proportion close to 1 represent 'cities' while those with a built-up proportion close to 0 represent the smallest possible town. Results are shown in Fig B1 (in levels) and B2 (in logs) in S1 Appendix. If Zipf's law holds, then we expect a downward sloping straight line in Fig B2 in S1 Appendix. As clear from the figures, we find absolutely no evidence of a Zipf's Law for partial buildup similar to what has been discussed for cities [34, 35].

In Maps 1a to 1d (Fig 1), we illustrate what the data looks like for a randomly chosen location in Ghana—the rural town of Yeji. Buildup is color coded, with white meaning no evidence of any building in the cell and black meaning high density buildup—i.e., 90% and above. Intermediate shades of blue and grey correspond to different levels of partial buildup. We see that in 1975 there is very little evidence of buildup – most of Map 1a is empty. But the small settlement of Yeji can be seen in the upper middle of the map. Over time, this initial settlement grows and expands geographically. It also spawns small satellite settlements at more or less random but fairly equally spaced intervals. New settlements also appear farther away from the initial settlement, often in the proximity of a road. But the initial settlement remains the most densely built-up area of the map throughout the 40 years period covered by the data, suggesting some form of preferential attachment/first-mover advantage. If we enter the geographical

**Table 2. Presence of roads in a cell.**

| Road quality | 1976 | 1986 | 2008 | 2014 |
|---|---|---|---|---|
| Class 1 road | 0.61% | 0.67% | | |
| Class 2 road | 0.83% | 0.98% | | |
| Class 3 road | 1.03% | 1.15% | | |
| Track | 0.55% | 0.60% | | |
| Unknown | 0.01% | 0.00% | | |
| Any road | 2.99% | 3.35% | | |
| Intersection | 0.03% | 0.04% | | |
| **Administrative road definition** | | | | |
| Highways | | | n.a. | 0.00% |
| Primary | | | 0.63% | 0.25% |
| Secondary | | | 0.36% | 1.05% |
| Tertiary | | | 0.82% | 1.66% |
| Local road | | | n.a. | 0.28% |
| Any road | | | 1.80% | 3.12% |
| Intersection | | | n.a. | 0.23% |

Notes: Number of observations: 3,880,380. Unknown roads in 1976 and 1986 denote a very small proportion of roads of unknown quality. Road quality in 1976 and 1986 does not align with administrative road definitions in 2008 and 2014, which themselves differ as well in spite of the partial similarity in names. Highways in 2014 include a single motorway/freeway from Accra to the nearby city of Tema and is present in 113 cells only. Averages for road availability in the 8 (N1) and 16 (N2) neighboring cells are not shown here to save space. They are essentially identical to the averages reported here since they are averages over essentially the same cells (except for small differences at the edges). Intersections cover all classes of road available in the data.

**Table 3. Time-invariant features of cells.**

| | | Average | Min | Max |
|---|---|---|---|---|
| | Latitude | 7.96 | 4.74 | 11.17 |
| | Longitude | -1.21 | -3.26 | 1.20 |
| | Presence of water body | 2.7% | | |
| | Presence of land border | 0.1% | | |
| | Presence of sea | 0.1% | | |
| | Number of cells | 3,853,273 | | |
| **Soil quality** | | | | |
| | Poor | 38.1% | | |
| | Moderate | 47.8% | | |
| | Good | 7.4% | | |
| | Hydromorphic | 1.7% | | |
| | Waterlogged/Steep terrain | 5.0% | | |
| **Climate** | | | | |
| | Moist or dry | 17.1% | | |
| | Sub-humid | 70.2% | | |
| | Humid | 12.8% | | |
| | No. cells (soil, climate) | 3,811,014 | | |

Notes: Latitude and longitude reported in decimal degrees. Water bodies include rivers, lakes and lagoons. Hydromorphic soils are those affected by water (e.g., river banks, tidal estuaries). Waterlogged soils and steep terrain are those considered unsuitable for agriculture or house construction. Of all the cells covering the area of Ghana, 27,107 are excluded because they are fully covered by water; most of this is Lake Volta.

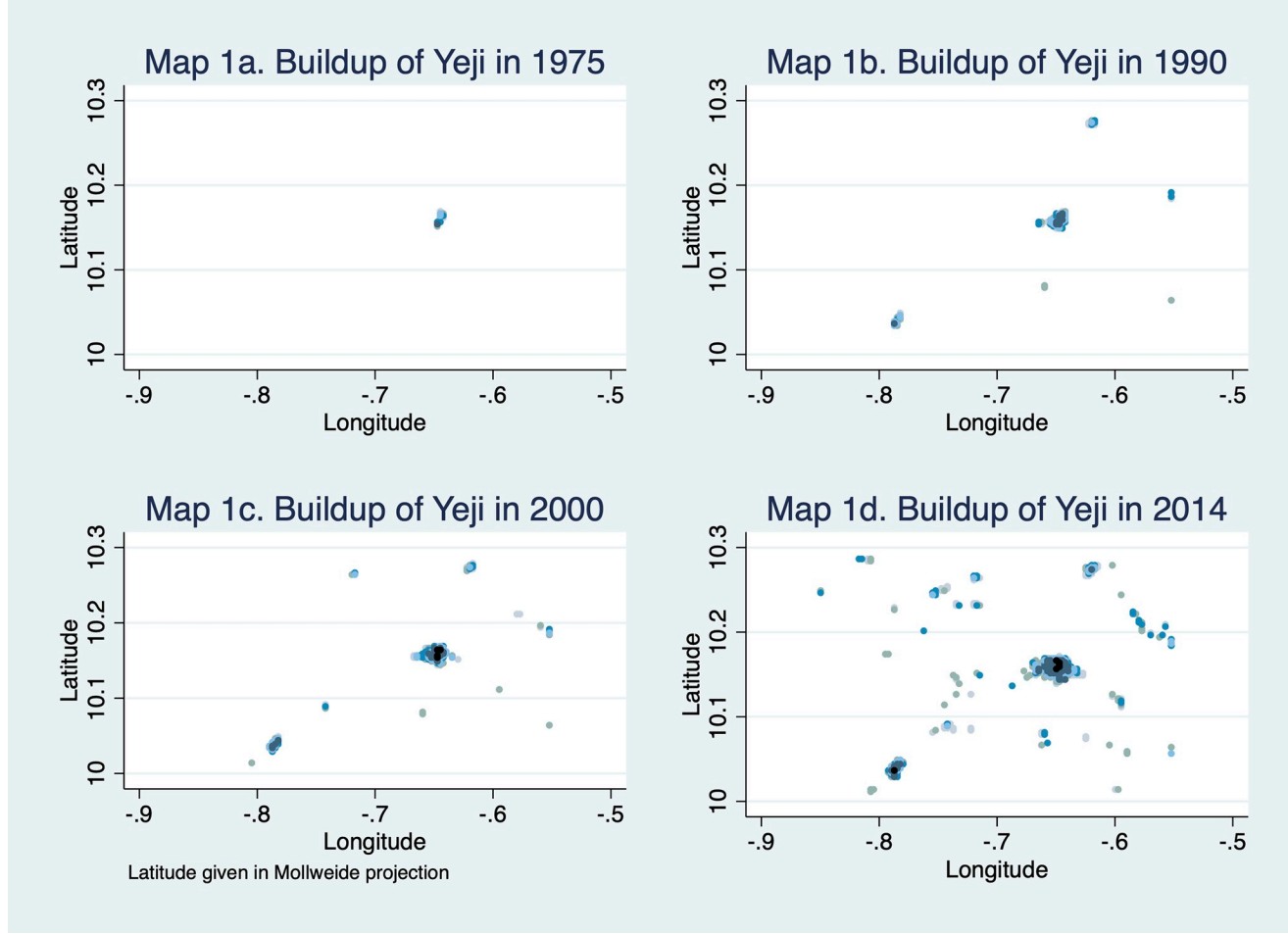

**Fig 1. Map 1.** Buildup of Yeji over time.

coordinates of Yeji (8˚ 05'56.4"N 0˚ 54'10.8"W) in Google Earth, it is immediately clear that the data contain a wealth of detail about buildup that comes to view under magnification. It also reveals the presence of the Volta lake nearby. Some of the roads shown in Google Earth correspond to 1975 roads. However, one road running to the west from Yeji seems no longer in use, even though its outline is still visible from the air. Similar map sequences for other parts of Ghana show a similar evolution, at least visually (Maps 2a to d (Fig 2) show the evolution of buildup in the coastal town of Takoradi). The rest of the paper is devoted to analyzing these patterns in a formal way.

Road information is presented in Table 2. For 1976 and 1986, the road information we have captures road quality, divided into four categories, plus a small 'quality unknown' category. At that time, most of the roads included in this list were unpaved. According to these data, the proportion of cells with at least one road was 3% in 1976 and 3.4% in 1986. The 2008 data use a more stringent road definition, according to which 1.8% of cells had at least one road. The 2014 data includes two new categories: highways, which at the time only include on short stretch of motorway/freeway between Accra and a nearby town; and local roads. The meaning of the other three categories also seems to have changed between 2008 and 2014, precluding a direct comparison. In the analysis we rely heavily on the 1976 data because it captures the pre-determined road infrastructure at the beginning of our study period. Averages for road

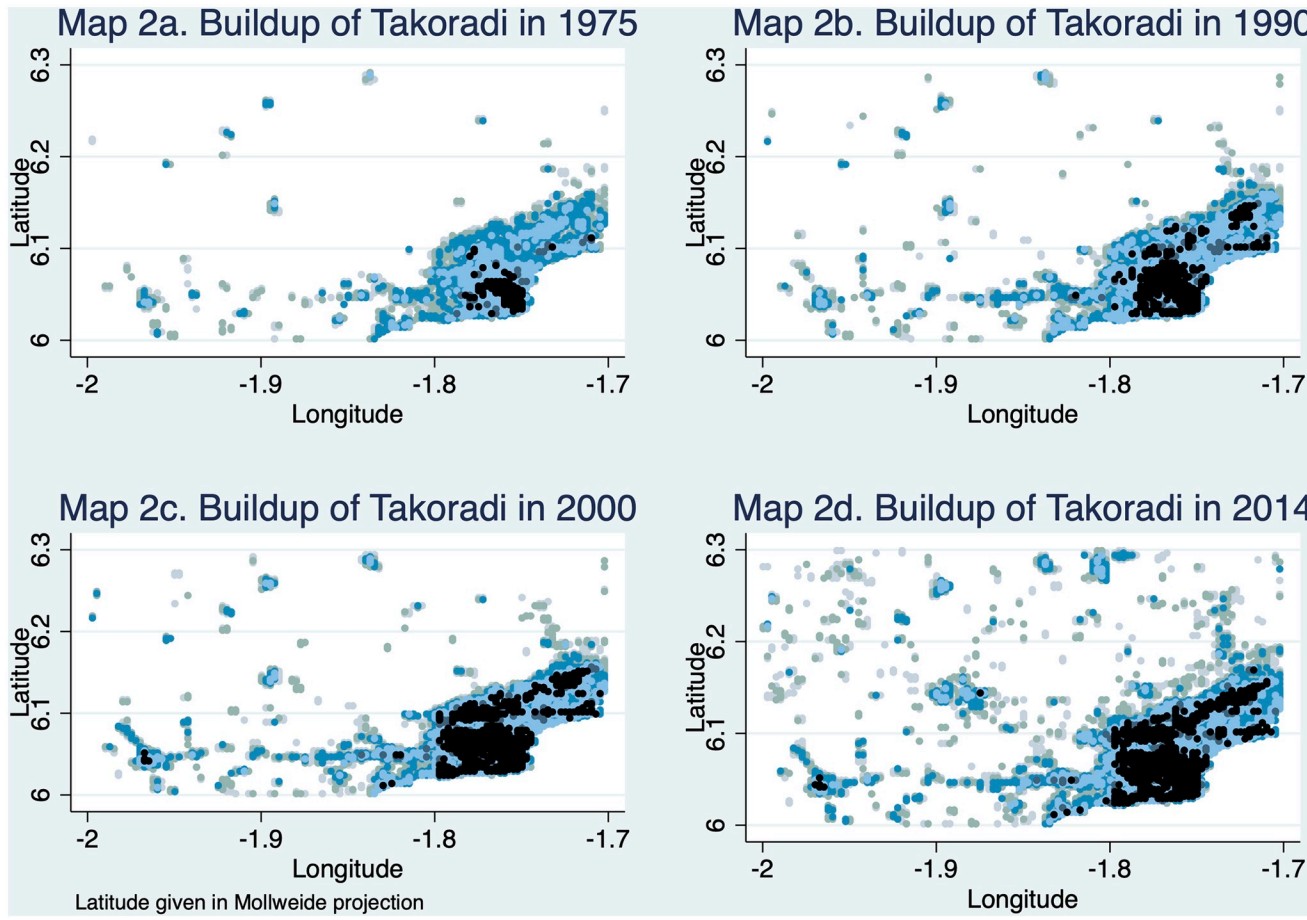

**Fig 2. Map 2.** Buildup of Takoradi over time.

availability in the 8 $n_1$ and 16 $n_2$ neighboring cells are not shown to save space but, by construction, they are identical to the cell averages reported here.

In Table 3 we summarize our main time-invariant variables. Longitude and latitude are presented in decimal degrees. They are used to measure the distance between cells in the subsequent analysis. A tiny fraction of cells are on the border with a neighboring country or on the seashore, but 2.7% are partially covered by a water body such as Lake Volta. The predominant soil quality is either poor or moderate, a situation that typifies much of Sub-Saharan Africa and reflects its tropical nature: heavy rainfall combined with high evapo-transpiration means considerable leaching of nutrients from the soil. Only 7.4% of soils are categorized as good. There is a small fraction of cells with hydroponic soils, which means they are located in river estuaries or flood plains. Some of these soils can be made fertile, but often at some cost in terms of infrastructure and equipment. Some 5% of cells have land unsuited for traditional agriculture either because they are waterlogged or too steep. Cultivating these soils requires investment in drainage or terracing. In terms of humidity, most of Ghana's land is classified as sub-humid, with some areas being either drier or more humid. Rainfall typically follows a North-South gradient in western Africa, with more rains along the coast than in the North. There are exceptions, however: coastal areas of Ghana tend to receive less rainfall than those located more inland. The humidity of the soil also depends on its texture. In Ghana, soils in the valley of the Volta river are classified as humid as they tend to retain moisture better.

**Table 4. Breakdown of built-up proportion by agro-ecological zones.**

| Soil Quality | | | | | Soil Humidity | | | No info | |
|---|---|---|---|---|---|---|---|---|---|
| Year | Poor | Moderate | Good | Hydromor. | Wtrlg/Steep | Moist/dry | Semi-humid | Humid | |
| 1975 | 0.55% | 0.30% | 0.21% | 0.41% | 0.09% | 0.21% | 0.47% | 0.15% | 3.56% |
| 1990 | 0.89% | 0.70% | 0.43% | 0.81% | 0.23% | 0.37% | 0.88% | 0.42% | 5.88% |
| 2000 | 1.31% | 1.08% | 0.66% | 1.07% | 0.29% | 0.50% | 1.32% | 0.66% | 8.09% |
| 2014 | 2.03% | 1.69% | 1.02% | 2.71% | 0.75% | 0.69% | 2.07% | 1.34% | 9.32% |
| N. obs. | 1,447,084 | 1,816,564 | 282,295 | 64,568 | 189,485 | 648,635 | 2,666,937 | 484,424 | 53,277 |

In Table 4 we break down $y_{it}$ by agro-ecological zone and proximity to borders and water. We find strong initial differences in buildup density across zones with, in 1975, more buildings (and hence a higher population density) in semi-humid areas and in areas with humid or hydromorphic soils. Over the study period, however, we observe dramatic changes in relative buildup densities: cells with initially the lowest buildup densities—i.e., areas with soils that are humid, hydromorphic, or waterlogged or steep—experience the fastest growth, possibly suggesting that investments were successful in making these soils or locations more productive. Reduction in onchocercosis—also known as river blindness—may have also attracted settlement in these areas.

Table 5 presents a similar breakdown of built-up areas by the presence or not of a road in the cell. Not surprisingly, we find a strong difference in buildup density between cells with and without a road. This difference falls over time however: comparing cells with and without a road in 1976, the difference falls from 8.5 more buildup in 1975 to 6.3 times in 2014. We also note that the new road definitions adopted in the 2014 data match heavily built-up areas much more closely than before, suggesting that this new road nomenclature puts more emphasis on residential streets [57] and lesson dirt roads serving rural communities.

In Table 6 we show buildup proportions in 'edge cells', that is, cells that overlap with a land border or a water body. We see that the presence of a water body is in general associated with a

**Table 5. Breakdown of built-up proportion by presence of a road.**

| Year | Road in 1976 | | Road in 1986 | | Road in 2008 | | Road in 2014 | |
|---|---|---|---|---|---|---|---|---|
| | No | Yes | No | Yes | No | Yes | No | Yes |
| 1975 | 0.35% | 2.94% | 0.34% | 3.01% | 0.37% | 3.53% | 0.22% | 6.84% |
| 1990 | 0.67% | 5.21% | 0.64% | 5.39% | 0.70% | 6.23% | 0.46% | 11.44% |
| 2000 | 1.01% | 7.15% | 0.97% | 7.44% | 1.05% | 8.61% | 0.74% | 14.96% |
| 2014 | 1.59% | 10.00% | 1.55% | 10.35% | 1.66% | 12.21% | 1.29% | 19.17% |
| N. obs. | 3,737,881 | 115,392 | 3,723,869 | 129,404 | 3,783,821 | 69,452 | 3,732,420 | 120,853 |

**Table 6. Breakdown of built-up proportion by boundary status.**

| Year | Presence on the cell of: | | | |
|---|---|---|---|---|
| | Water body | Land border | Sea | None |
| 1975 | 0.01% | 0.27% | 30.75% | 0.42% |
| 1990 | 0.03% | 0.52% | 49.73% | 0.80% |
| 2000 | 0.06% | 0.70% | 60.40% | 1.19% |
| 2014 | 0.20% | 1.11% | 61.87% | 1.86% |
| N. obs. | 105,035 | 4,041 | 1,978 | 3,742,297 |

much lower buildup density. This may be because the largest inner body of water in Ghana is a dam lake and hence probably has a water level that varies over time depending on annual rainfall: people don't want to live right next to it. Land borders also have a lower density than no-edge cells, while cells near the sea have a much higher buildup density, probably reflecting the presence of towns and cities along the coast, but also of fishing communities that launch their fishing boats directly from the beach. We also observe a much faster buildup over time in cells near inner water bodies, probably reflecting the fact that Lake Volta was only completed in 1965 and it took time for people to relocate along its shore.

## Buildup frequency distribution

Next, we look in more detail at the frequency distribution of buildup for cells that are neither empty of buildings or fully occupied by them. Fig 3 shows this distribution for the four data years we have. It is immediately clear that the frequency distribution of partially built cells remains constant over time, even if the proportion of unbuilt cells falls while the proportion of fully built-up cells increases. We then break down this distribution by cells that are located near built-up areas and cells that are not, i.e, by whether $y_{it}^{n_2} > 0$ or $= 0$. Unsurprisingly perhaps, we find that isolated built-up cells only have a small fraction of built-up area: half of them are less than 4% built up, which is equivalent to having at most a $50m^2$ (550 square feet) structure on them; three quarters of them have structures covering at most $83m^2$ or 915 square feet. In contrast, half of built-up cells in the vicinity of other built-up cells have structures covering $106m^2$ or more—with a fairly even distribution of $y_{it}$ across the entire spectrum of built-up proportion, a finding in line with the idea that these cells correspond to the edges of towns and cities. This pattern suggests that population growth at the aggregate level is not associated with a densification of all areas simultaneously. Rather, what we observe is a fall in the

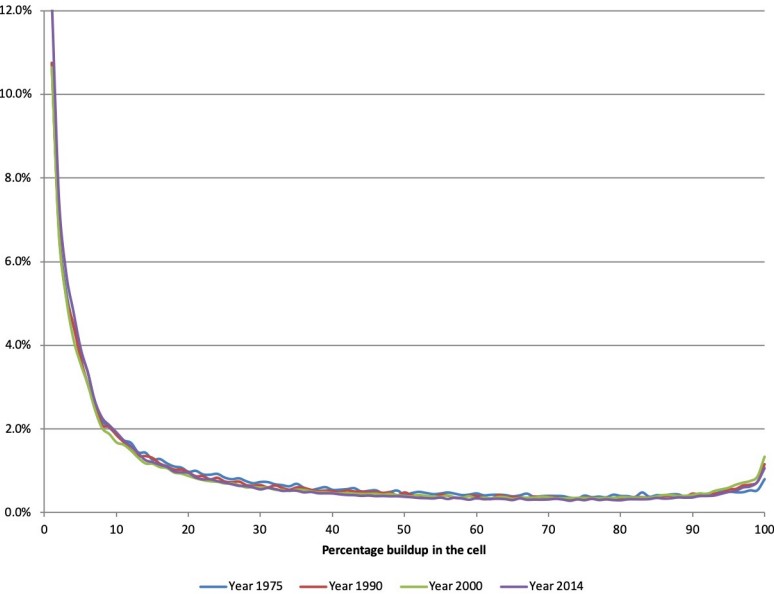

**Fig 3. After dropping unbuilt cells (which constitute more than 90% of cells in all years) and the fully built-up cells (which are discussed separately in the text), Fig 3 plots four histograms of the frequency distribution of partially built-up cells by their built-up proportion.** The horizontal axis shows the built-up proportion of the cell. The vertical axis shows the proportion of cells in each buildup percentage interval. There is one histogram for each year. To facilitate viewing, each of the four histograms is represented as a colored line and the four histograms are overlaid on top of each other. This is done to facilitate a visual comparison of the frequency distribution across years.

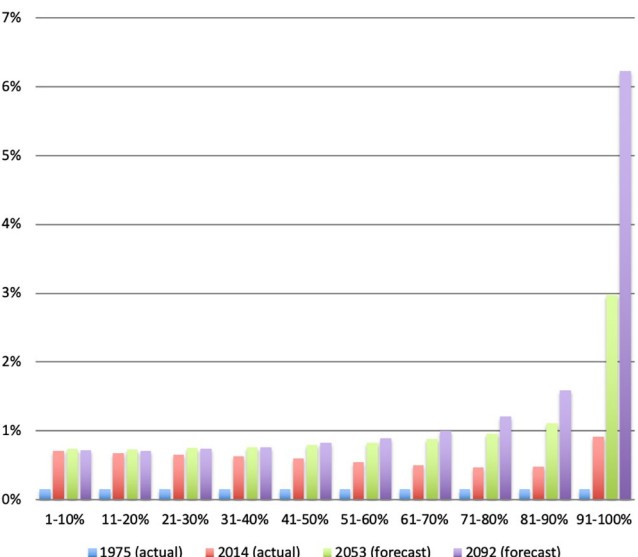

**Fig 4. Fig 4 shows a stacked histogram of the frequency distribution of built-up proportion divided into deciles of buildup proportion in 1975.** Fully built cells are included in the top decile. Since an overwhelming proportion of cells are unbuilt in all years, they are omitted from the Figure for the sake of clarity. The 1975 and 2014 values are taken directly from the data. The 2053 and 2092 forecasts are obtained as follows. We first form the decile transition matrix using data from 1975 and 2014 (a 39-years interval). We then project the built-up frequency distribution from 2014 to 2053 and 2092 as two iterations of the transition matrix. The resulting forecasts should be understood as predictions of what the distribution of buildup will look like if the transition matrix remains unchanged. Similar forecasts are obtained if we use transition matrices constructed using different data years. The forecasts do not specify which cells will be built.

proportion of empty cells, an increase in the proportion of densely built cells, and a more or less constant distribution of partially built cells.

To investigate this issue further, we plot the forecasts based on the transition matrix $M$ in Fig 4, together with the actual built-up proportions in 1975 and 2014. The histogram confirms that the evolution of built-up proportions in Ghana is characterized by a transfer from unbuilt to fully built, with partially built-up cells serving a transitory role. Over the foreseeable future, the calculations forecast 89.5% unbuilt cells circa 2050 and 85% unbuilt cells circa 2100. At the same time, the proportion of densely built areas (90% built-up or more) is forecasted to go from 0.9% in 2014 to 3% in 2053 and 6.2% in 2092. No equivalent increase is predicted for partially built cells. In other words, the additional population will not spread itself evenly on the land. Rather, it will congregate in dense towns and cities. Assuming that the historical rate of population growth continues, this represents more than a tripling of dense urban areas in about 40 years, and a multiplication by 7 by the end of the century. Similar forecast are obtained if we construct the transition matrix using only 1976 and 1986, or 2000 and 2014: the resulting transition matrices are very similar, and the forecasts as well. Although our forecasts apply to buildings only, they imply a dramatic change in the distribution of population across the country, with an increased concentration in towns and cities without much of a densification of rural areas. To illustrate the magnitude of these changes, an equivalent increase in the population of Accra would imply an increase from 2.2 million in 2014 to 7.3 million people by 2053 and more than 15 million by the end of the century. These are not small changes.

## Roads

In Fig 5a and 5b, we repeat the exercise separately for cells that were close to a road in 1976 and those that were not. We define being close to a road as having at least one of the five

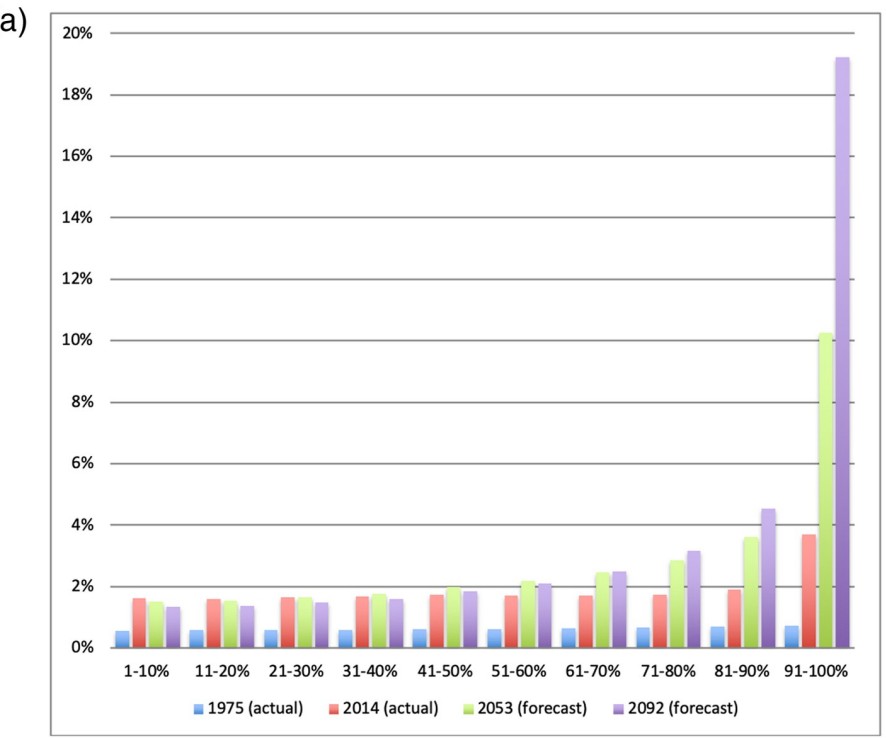

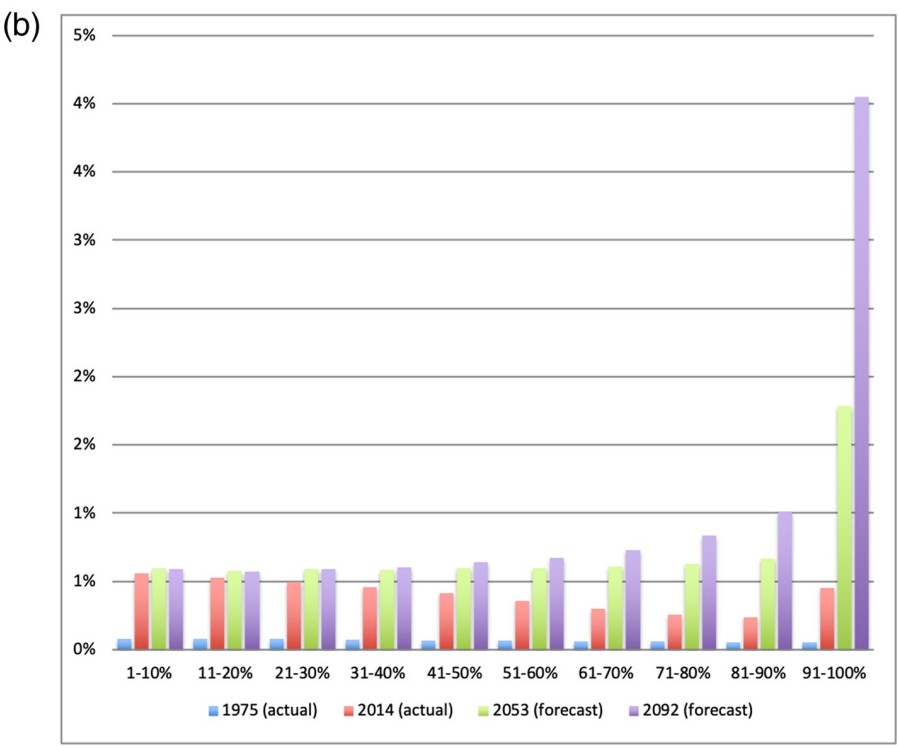

**Fig 5. The Figures show a stacked histogram of the frequency distribution of built-up proportions divided into 1975 deciles.** In Fig 5a only those cells with a road in 1976 are used in the histograms. Fig 5b does the same thing for cells without a road in 1976. In both Figures, fully built cells are included in the top decile. Since an overwhelming proportion of cells are unbuilt in all years, they are omitted from the Figures for the sake of clarity. The 1975 and 2014

values are taken directly from the data. The 2053 and 2092 forecasts are obtained as follows. We first form the decile transition matrix using data from 1975 and 2014, only using cells with a road in 1976 (for Fig 5a) or those without a road in 1976 (for Fig 5b). We then separately project the built-up frequency distribution from 2014 to 2053 and 2092 as two iterations of the transition matrix. The resulting forecasts should be understood as predictions of what the distribution of buildup will look like in 1976 if the transition matrices remain unchanged. Similar forecasts are obtained if we use transition matrices constructed using different data years. The forecast does not specify which cells will be built.

categories of roads documented in 1976 either in the cell itself, or in its neighborhood $n_1$ or $n_2$. With this definition, 14.24% of cells at most 1.5km from a road in 1976. The year 1976 is chosen because it is close to the starting point of our data. Hence the transition matrices that are constructed on that basis can be seen as predictions based on road conditions existing in 1976. Results show a stark contrast between cells near road or not. Cells near a 1976 road were already less likely to be unbuilt in 1975: 93.7% vs. 99.3% for cells not near a road. This gap increased to 81% vs. 96% in 2014 and is predicted to grow to 61% vs. 90% by the end of the century. Furthermore, the distribution of built-up areas also varies greatly depending on the presence of a road in 1976. In 1975, only 0.05% of cells not near a road had a value of $y_{it}$ in excess of 90%, compared to 0.73% of those near a road. By 2014 the difference was 0.46% vs. 3.71%. By the end of the century, the forecasted difference is 4% vs. 19%. Put differently, by the end of the century nearly 19% of the area within 1.5km of a 1976 road will be nearly fully built up. Of course new roads will be built that can affect that, a point we revisit below. But the main message here is that the early presence of a road is predicted to have long lasting effects on urbanization patterns. Fig 3a and 3b also show that while proximity to a road affects the probability of subsequent buildup, it does not markedly change the distribution of partially built-up cells: while the levels are different between the two figures, the shape of the distribution is similar. This is true even though each transition matrix is estimated from two distinct sets of observations.

Next, we expand on the above analysis to investigate the effect of adding or removing a nearby road. We use the data from 1976 and 1986 since they are comparable. We define four categories: cells not close to a road in either 1976 or 1986; cells close to a 1976 road that no longer existed in 1986; cells close to a road that did not exist in 1976 but was present in 1986; and cells close to a road that existed both in 1976 and 1986. Abandoning a road may signal that the road was no longer economically useful which, if true, should predict less growth and less subsequent buildup. In contrast roads that remained in existence across both years should be correlated with a pre-existing and persistent economic usefulness. Hence, we expect them to be most associated with economic activity and thus most likely to attract population and buildup. Roads that were added in 1986 should occupy an intermediate position: they were not initially deemed useful but were added later, perhaps when sufficient funds became available to add less essential roads. For the analysis, we construct four transition matrices from the 1975–2014 data on these four categories of cells. We then predict buildup in 2053 and 2092 as before. To save space, we present in Fig 6a and 6b only the results for unbuilt and fully built cells, respectively. Indeed, as noted earlier, the actual and forecasted distributions of partially build cells are fairly stable across time. We see that, as anticipated, cells near a road are predicted to receive a lot more buildup over time. The buildup forecast is particular strong for locations that had a road in 1986. Whether or not they also had a road in 1976 matters less, contrary to what we expected. Cells near an abandoned road do less well, but still better than cells with no road in either years.

(a)

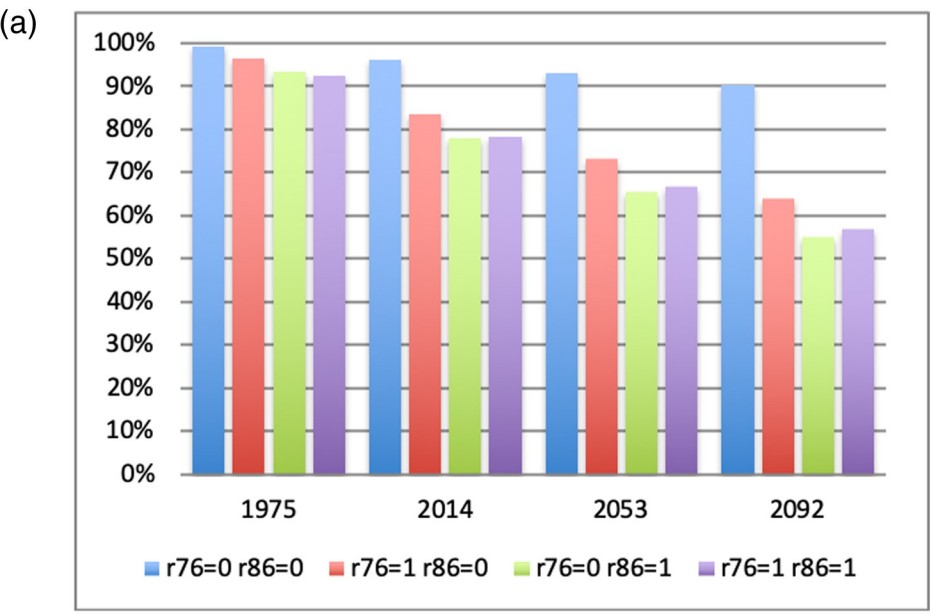

(b)

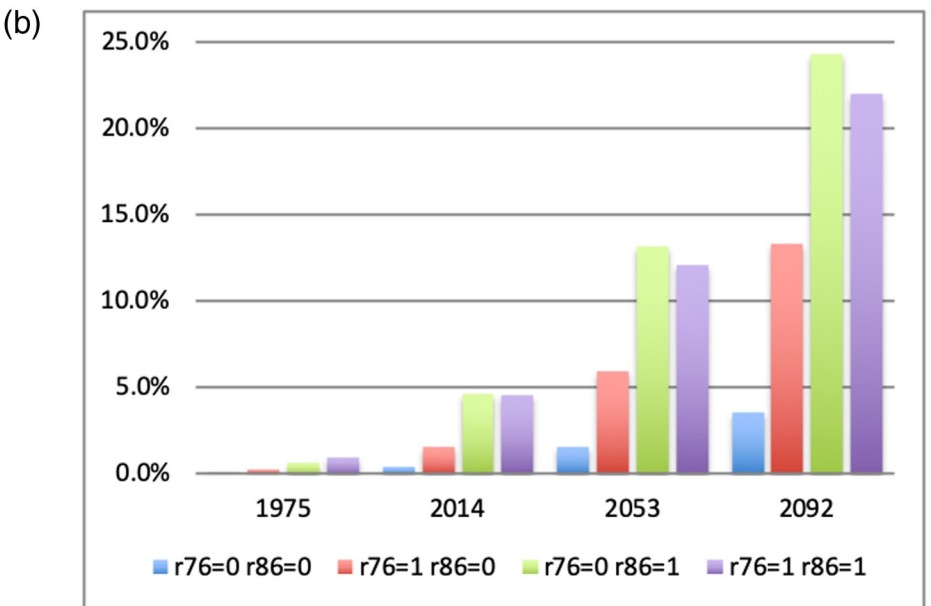

**Fig 6. Fig 6a shows the percent of unbuilt cells in various years, conditional on whether the cell had a road in 1976 (r76 = 1) and a road in 1986 (r86 = 1).** Some cells with a road in 1976 no longer have one in 1986 (e.g., because the road was not maintained or was moved). Fig 6b shows the percent of fully built cells in various years, also depending on whether the cell had a road in 1976 (r76 = 1) and a road in 1986 (r86 = 1). Values for 1975 and 2014 are take from the data. Values for 2053 and 2092 are forecasts obtained in the same manner as for Fig 5a and 5b, except that we now use four different transition matrices and four 2014 buildup vectors from which to iterate.

## Agglomeration

Having discussed roads, we now turn to agglomeration effects. Table 7 reports the results from the estimation of Eq (2). The results, shown in Table 7, are very stable across data years: we note a strong contemporaneous correlation between $y_{it}$ and $y_{it}^{n_1}$ but, conditional on $y_{it}^{n_1}$, the

**Table 7. Correlation in buildup between neighboring cells.**

| | 1975 | | 1990 | | 2000 | | 2014 | |
|---|---|---|---|---|---|---|---|---|
| | Coef. | Std.err. | Coef. | Std.err. | Coef. | Std.err. | Coef. | Std.err. |
| y(n1) | 1.3102 | 0.0008 | 1.3761 | 0.0007 | 1.3953 | 0.0007 | 1.3708 | 0.0007 |
| y(n2) | -0.3237 | 0.0008 | -0.3888 | 0.0008 | -0.4060 | 0.0007 | -0.3807 | 0.0008 |
| Intercept | 0.0000 | 0.0000 | 0.0001 | 0.0000 | 0.0001 | 0.0000 | 0.0002 | 0.0000 |
| R-square | 0.7952 | | 0.8402 | | 0.8666 | | 0.8507 | |
| N. obs. | 3,853,228 | | 3,853,228 | | 3,853,228 | | 3,853,228 | |

Notes: the dependent variable is y. There are no controls included.

correlation with $y_{it}^{n_2}$ is negative. Put differently, if we observe high buildup in $n_1$ and $n_2$, we would predict a lower buildup $y_{it}$ than if we only observed high buildup in $n_1$ alone. This potentially suggest that buildups on either side of $n_1$ are substitutes: if happens in $n_2$, it is less likely to occur in cell $i$.

Next, we investigate whether cells in a built-up neighborhood subsequently experience more buildup. Results are presented in Table 8 for the six separate regressions corresponding to specification given in Eq (3). Unsurprisingly, they show a lot of persistence in buildup: the estimated coefficient $\beta_1$ is large in all regressions, indicating that past buildup is a strong predictor of future buildup. This predictive power, however, falls with the time gap $s$: the more distant in the past is $y_{it-s}$, the smaller the value of $\beta_1$, and the smaller is the precision of the prediction, i.e., the $R^2$. We also notice that, when the time interval $s$ is short, the additional

**Table 8. Correlation in buildup between neighboring cells over time.**

| Regressors in 1975 | 1990 | | 2000 | | 2014 | |
|---|---|---|---|---|---|---|
| | coef. | std.err. | coef. | std.err. | coef. | std.err. |
| y | 0.8691 | 0.0008 | 0.7273 | 0.0013 | 0.6123 | 0.0018 |
| y(n1) | 0.3420 | 0.0016 | 0.4390 | 0.0025 | 0.5221 | 0.0036 |
| y(n2) | 0.1605 | 0.0013 | 0.4304 | 0.0021 | 0.5568 | 0.0030 |
| intercept | 0.0022 | 0.0000 | 0.0051 | 0.0000 | 0.0113 | 0.0000 |
| R-square | 0.7453 | | 0.5945 | | 0.4352 | |
| **Regressors in 1990** | | | **2000** | | **2014** | |
| | | | coef. | std.err. | coef. | std.err. |
| y | | | 0.8961 | 0.0007 | 0.7575 | 0.0012 |
| y(n1) | | | 0.1839 | 0.0013 | 0.3688 | 0.0024 |
| y(n2) | | | 0.1025 | 0.0011 | 0.1477 | 0.0019 |
| intercept | | | 0.0024 | 0.0000 | 0.0082 | 0.0000 |
| R-square | | | 0.812 | | 0.6029 | |
| **Regressors in 1990** | | | | | **2014** | |
| | | | | | coef. | std.err. |
| y | | | | | 0.8395 | 0.0008 |
| y(n1) | | | | | 0.2505 | 0.0016 |
| y(n2) | | | | | 0.0252 | 0.0012 |
| intercept | | | | | 0.0052 | 0.0000 |
| R-square | | | | | 0.7758 | |

Notes: each panel is a separate regression. The total number of observations is 3,853,228 in each regression.

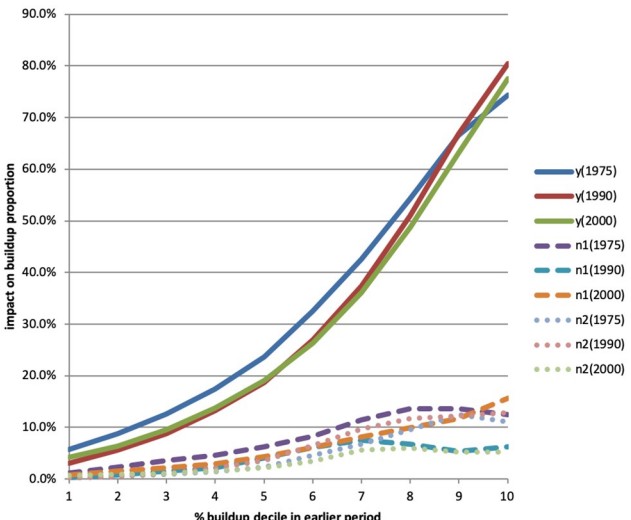

**Fig 7. Fig 7 summarizes the coefficients from three separate regressions.** The dependent variable in the three regressions is the proportion of a cell that is built up in 1990, 2000, and 2014 respectively. Each regression includes lagged values of buildup in the cell and in the neighboring cells n1 and n2. The lagged values are for 1975, 1990, and 2000 respectively. Each of the three lagged buildup variables appears in the regression as a series of 10 dummies, one per buildup decile. The Figure shows the estimated decile coefficients in each of the three regressions for (1) buildup in the cell itself [lines y(1975), y(1990) and y(2000)]; buildup in neighboring cells n1 [lines n1(1975), n1(1990), and n1 (2000)]; and buildup in neighboring cells n2 [lines n2(1975), n2(1990), n2(2000)]. Each regression also includes the following time invariant regressors: longitude, latitude, and their product [three variables]; soil humidity and soil quality dummies [six variables]; and dummies for the presence of a water body, a land border, or the sea in the cell, the neighboring cells n1 and the neighboring cells n2, respectively [9 variables]. The number of observations in each regression is just shy of 3.8 million. R-squares are 0.799, 0.845 and 0.796, respectively. All coefficients shown here are significant at the 1% or better.

predictive power of $y_{it}^{n_1}$ and $y_{it}^{n_2}$ is relatively small. As the time interval increases, the coefficients—and thus the predictive power—of $y_{it}^{n_1}$ and $y_{it}^{n_2}$ increases and, unlike in Table 7, it is positive for both. This is consistent with a process by which much of the diffusion of buildup to neighboring cells occurs over a relatively short time interval of less than 10 to 15 years. Once this diffusion has begun, own past buildup takes over the predictive power of buildup in neighboring cells. This finding is suggestive of some form of hysteresis: once a cell has started to build up, it tends to remain that way, irrespective of neighboring effects.

This conjecture about persistence is confirmed when we estimate a nonparametric version of Eq (3) whose specification is given in Eq (4). Coefficient estimates for the buildup variables of interest are presented in graphical form in Fig 7. Results by and large confirm earlier findings from Table 8. They also provide some evidence of a stronger lag effect for higher deciles of $y_{it-s}$, suggesting more persistence for high buildup. Interestingly, we see that the lagged effect of buildup in neighboring cells tends to taper off at high buildup values, consistent with some form of substitution effect.

Next, we explore whether the agglomeration and persistence patterns found above hold true for different road types. We estimate the regression specification given in Eq (5). We focus our attention on outcome years 1990 and 2000 because for each of them we have comparable road data approximately 14 years earlier, i.e., in 1976 and 1986. Road data for 2008 and 2014 are not comparable to either of these or to each other. Coefficient estimates for $\gamma_1$, $\gamma_2$ and $\gamma_3$ are presented in Fig 8, for each of the four road categories and each of the two outcome years, 1990 and 2000. Coefficient $\gamma_1$ gives the predicted % point increase in buildup if there is a road in the cell 14 years before. Coefficients $\gamma_2$ and $\gamma_3$ give the predicted % point increase in

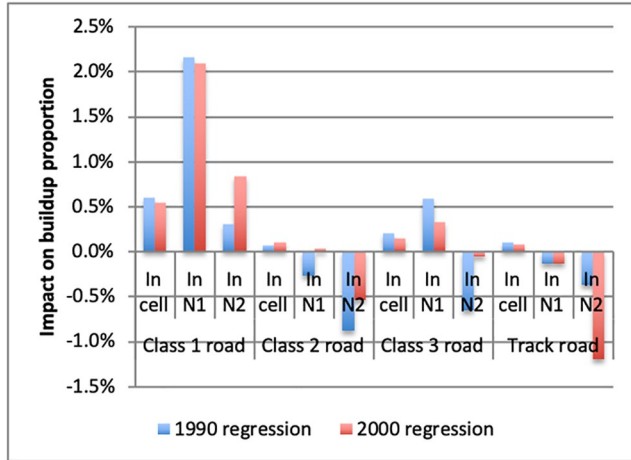

**Fig 8. Fig 8 summarizes the coefficients from two separate regressions.** The dependent variable in the two regressions is the proportion of a cell that is built up 1990 and 2000, respectively. The Figure shows the coefficients of the presence of a road of a certain class (i.e., category) in the cell, the neighborhood N1 and the neighborhood N2. The road quality ordering is: class 1 (top category), class 2, class 3, and track (bottom category). In the 1990 regression, the road data refers to 1976; in the 2000 regression, the road data refers to 1986. Each of the regressions also includes all the regressors that were present in those for Fig 7. More precisely, each regression includes lagged values of buildup in the cell and in the neighboring cells n1 and n2. The lagged values are for 1975 and 1990, respectively. Each of the three lagged buildup variables appears in the regression as a series of 10 dummies, one per buildup decile. Each regression also includes the following time invariant regressors: longitude, latitude, and their product [three variables]; soil humidity and soil quality dummies [six variables]; and dummies for the presence of a water body, a land border, or the sea in the cell, the neighboring cells n1 and the neighboring cells n2, respectively [9 variables]. The number of observations in each regression is just shy of 3.8 million. R-squares are 0.796 and 0.796, respectively. All coefficients shown here are significant at the 1% or better.

buildup if neighborhoods $n_1$ and $n_2$, respectively, go from no cell with a road to all cells with a road. We note a lot of similarity between the two regressions in terms of relative magnitude of the coefficients: the presence of a class 1 road in the cell or nearby predicts more buildup 14 years later. For the other three categories, the presence of a road in the cell predicts slightly higher buildup later on, but the presence of a road in the more distant neighborhood $n_2$ has a consistently negative effect on subsequent buildup predictions. A similar negative coefficient is observed in some regressions for neighborhood $n_1$ as well. These findings are again consistent with the existence of a substitution effect in buildup: if the road is in a neighboring cell but not in my cell, my cell is likely to be built-up a decade and a half later—an outcome we would expect if local inhabitants prefer to locate immediately near these smaller roads. When we estimate regression (5) with 2014 buildup $y_{it}$ and the 2008 road data, we find that all the road coefficients but one are positive. The only exception is for the presence of a primary road in the cell itself, which by 2014 may have too much traffic to be an attractive location. The 2008 road data is completely different from the earlier data, however, with very little correlation between the two, thereby making comparisons problematic. The 2014 road data is strongly correlated with buildup in that year, but since it is contemporary to our last data year, it cannot be regarded as a useful predictor.

We also estimate a different version of regression (5) setting $t - s = 1975$ for lagged buildup and $t - s = 1976$ for the road data. By regressing each of the subsequent data years 1990, 2000 and 2014 on these initial buildup and road conditions, we examine the delayed predictive effect of buildup and roads over time. Fig 9 summarizes the buildup coefficients in a way similar to Fig 6, the main difference being that now all regressions use the same 1975 buildup data as regressors. The Figure shows clearly that, over time, the predictive effect of past buildup

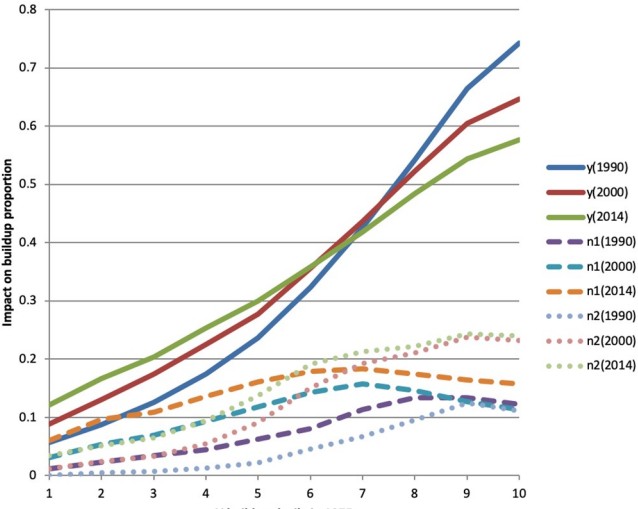

**Fig 9. Fig 9 summarizes the coefficients from three separate regressions.** The dependent variable in the three regressions is the proportion of a cell that is built up in 1990, 2000, and 2014 respectively. Each regression includes the 1975 values of buildup in the cell and in the neighboring cells n1 and n2. Each of the three 1975 buildup variables appears in the regression as a series of 10 dummies, one per buildup decile. The Figure shows the estimated decile coefficients in each of the three regressions for (1) 1975 buildup in the cell [lines y(1990), y(2000) and y(2014)]; 1975 buildup in neighboring cells n1 [lines n1(1990), n1(2000), and n1(2014)]; and 975 buildup in neighboring cells n2 [lines n2(1990), n2(2000), n2(2014)]. Each regression also includes all the road presence dummies for 1976 (see Fig 10) and the following time invariant regressors: longitude, latitude, and their product [three variables]; soil humidity and soil quality dummies [six variables]; and dummies for the presence of a water body, a land border, or the sea in the cell, the neighboring cells n1 and the neighboring cells n2, respectively [9 variables]. The number of observations in each regression is just shy of 3.8 million. R-squares are 0.799, 0.711 and 0.582, respectively. All coefficients shown here are significant at the 1% or better.

changes. For buildup in neighborhoods $n_1$ and $n_2$, we see a systematic increase in buildup predictions as time passes. This is consistent with some form or preferential attachment—the presence of buildings in the past attracts new buildings later—possibly due to agglomeration externalities. Past buildup in the cell itself varies depending on the original buildup intensity: for heavily built-up cells in 1975, the own-cell predictive effect falls over time. In contrast, cells with less buildup in 1975 see their buildup predictions increase over time. The fall in predictions for heavily built-up cells may suggest some form of congestion, unless it is compensated by buildup in neighboring cells, which makes it harder to move to a nearby cell.

Fig 10 does the same thing for coefficient estimates for $\gamma_1$, $\gamma_2$ and $\gamma_3$. The interpretation of the coefficients is similar to that in Fig 8, except that we are now comparing the evolution of buildup predictions over time, using roads in 1976 as predictor. We immediately see that the predicted effect of class 1 roads increases massively over time both for the presence of a road in the cell, as well as for road density in neighboring cells. The other three road categories produced similar results: we observe an increasing positive effect of roads in the cell and in the immediate neighborhood $n_1$ on buildup predictions, while road presence in neighborhood $n_2$ has a negative effect that is initially stronger and falls later on. This is consistent with an initial substitution effect that starts to reverse itself as population density increases and moving to a neighboring cell to get near the road becomes harder.

## Spatial covariogram

So far we have been looking at agglomeration effects at fairly close range—i.e., up to 1.5km at most. We found mostly evidence of agglomeration effects, except occasionally when we

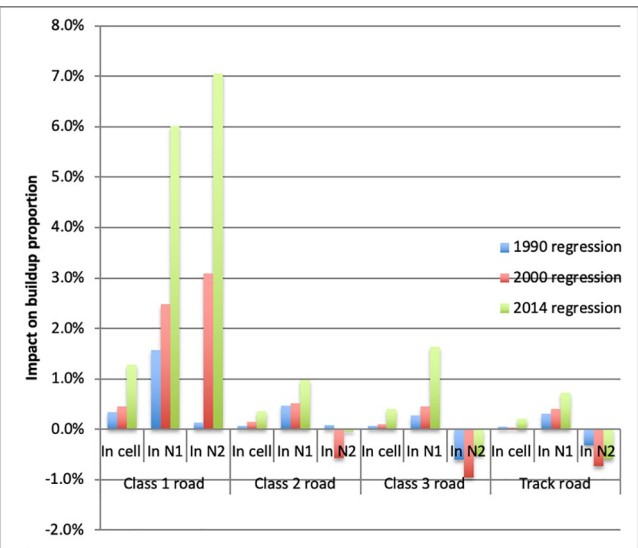

**Fig 10. Fig 10 summarizes other coefficients from three separate regressions presented in Fig 9.** The dependent variable in the three regressions is the proportion of a cell that is built up in 1990, 2000, and 2014 respectively. The Figure shows the coefficients of the presence in 1976 of a road of a certain class in the cell, the neighborhood N1 and the neighborhood N2. The road quality ordering is: class 1 (top category), class 2, class 3, and track (bottom category). Each regression includes the 1975 values of buildup in the cell and in the neighboring cells n1 and n2. Each of the three 1975 buildup variables appears in the regression as a series of 10 dummies, one per buildup decile (see Fig 9). Each regression also includes the following time invariant regressors: longitude, latitude, and their product [three variables]; soil humidity and soil quality dummies [six variables]; and dummies for the presence of a water body, a land border, or the sea in the cell, the neighboring cells n1 and the neighboring cells n2, respectively [9 variables]. The number of observations in each regression is just shy of 3.8 million. R-squares are 0.799, 0.711 and 0.582, respectively. All coefficients shown here are significant at the 1% or better.

uncovered patterns suggestive of spatial substitution effects. We now explore how buildup areas evolved over a much broader range of distances. This is done by estimating spatial covariograms as described in the empirical strategy section (2.3.3). First, we present the results when Ghana is divided into 856 equal sized tiles resulting in maximum distance of about 26km within each tile. The results are shown in Fig 11 for all four years combined. We see that, since the proportion of built-up cells increases over time, the proportion of built-up cell pairs not surprisingly increases as well. Otherwise the spatial pattern is similar across all years: cells located close to a built-up cell are more likely to be built up as well: $E[z_{it} z_{jt}]$ falls with distance. The correlation is particularly strong at short distances of up to 3 km, but the pattern is present across the entire distance distribution and, given how tight the confidence intervals are, differences in spatial correlation by distance are significant up to distances of at least 20km. Beyond 20 km, the number of observations in distance bins $N_{bm}$ drops rapidly, confidence intervals increase in size, and the covariogram becomes unreliable, a point we revisit below.

We repeat the exercise separately for cell pairs both located near a road in 1976, and other cell pairs. Proximity to a road is defined as having a road in the cell itself or in neighborhoods $n_1$ or $n_2$—i.e., as having $r_{it_0}$, $r_{it_0}^{n_1}$, or $r_{it_0}^{n_2} > 0$ for $t_0 = 1976$. The resulting spatial covariogram is shown in Fig 12 for all four years of data we have. The general pattern is similar to that observed for all cell pairs, except that the proportion of built-up cells is higher near roads, and thus there are more built-up cell *pairs* near roads. We also observe the sharply higher spatial correlation at close distances of around 2–3 km. But the spatial correlation in buildup falls less fast near roads than away from roads. This is consistent with our earlier finding that proximity to a road predicts higher buildup independently from proximity to a pre-existing

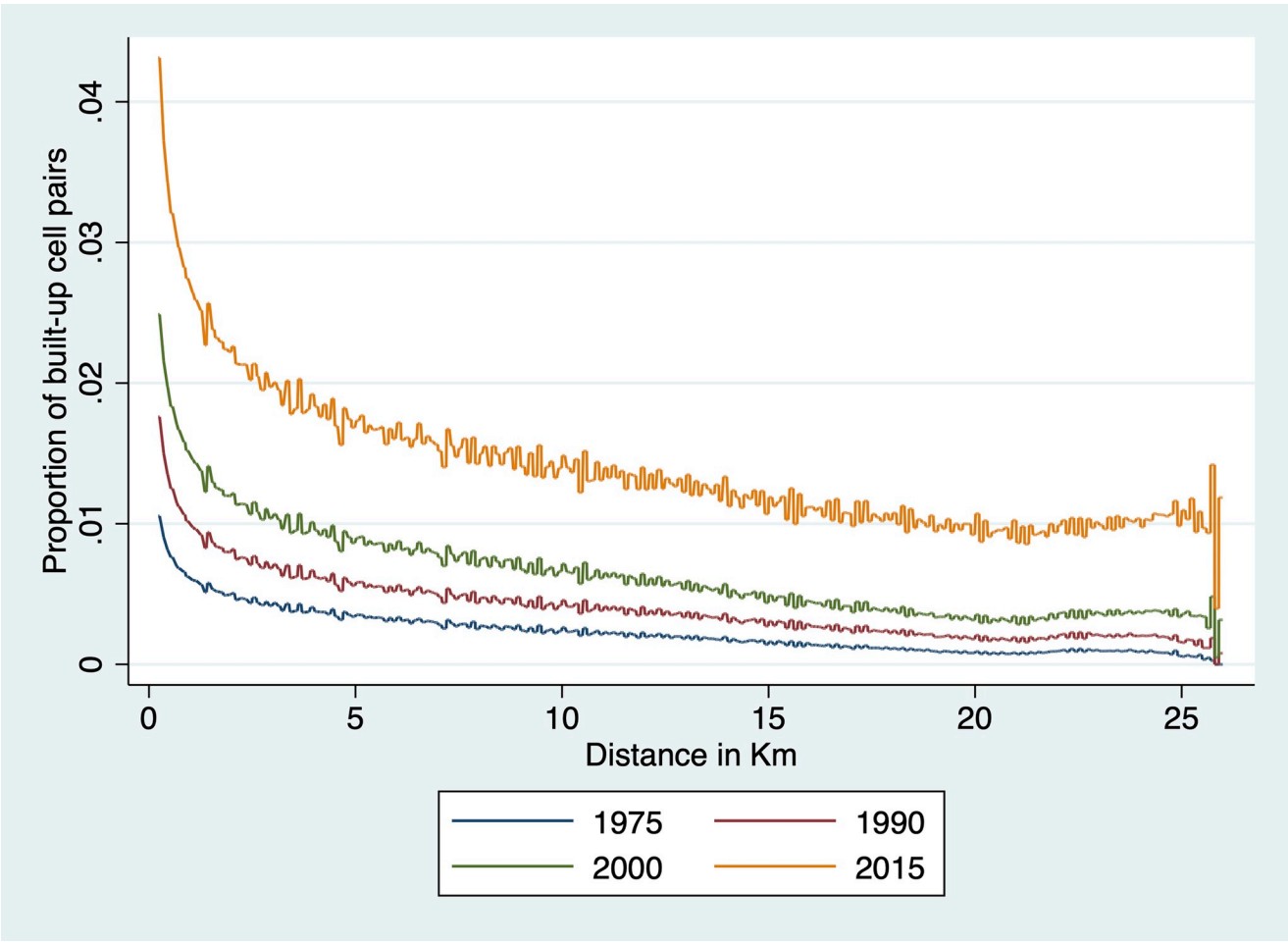

**Fig 11. Fig 11 shows, for each distance value within small tiles, the proportion of cell pairs that are both built-up.** For instance, a value of 2% at distance 4Km means that pairs with two built-up cells make up 2% of all the cell pairs distant from each other by 4Km. Four correlograms are displayed, one for each of the years for which we have buildup data.

agglomeration, and this effect is distinct from the fact that settlements often are near roads to start with. The spatial covariogram for cell pairs that are not close to roads is very similar to that reported in Fig 11 and is not shown here to save space.

Contrary to what is predicted by the von Thunen [31] model subsequently expanded by Chistaller [32] and Isard [33], we find no strong evidence of a 'bump' in the covariogram at any specific distance. If farm buildings tended to be spaced equally so as to locate in the middle of their fields, we would expect a spike in the covariogram at the relevant distance. For instance, if each farming households occupies, say, 20 Ha of cultivated land, fallows and pastures, we would expect farms to be, on average, 500 meters from each other. This would more or less correspond to one farm per cell, and thus a flat spatial covariogram between farm buildings. We also would not expect to find a buildup concentration along roads—which is the contrary of what we find. In contrast, if farmers live in villages of, say, 100 households needing each 20 Ha of combined cultivated land, fallows and pastures, each village would require an area of 2,000 Ha, which would imply a spacing of villages each 5 km. If this were the case, we would expect a spike in the spatial covariogram at 5km: the presence of village buildings at distance 0 would predict village buildings 5 km away. We do not find evidence consistent with

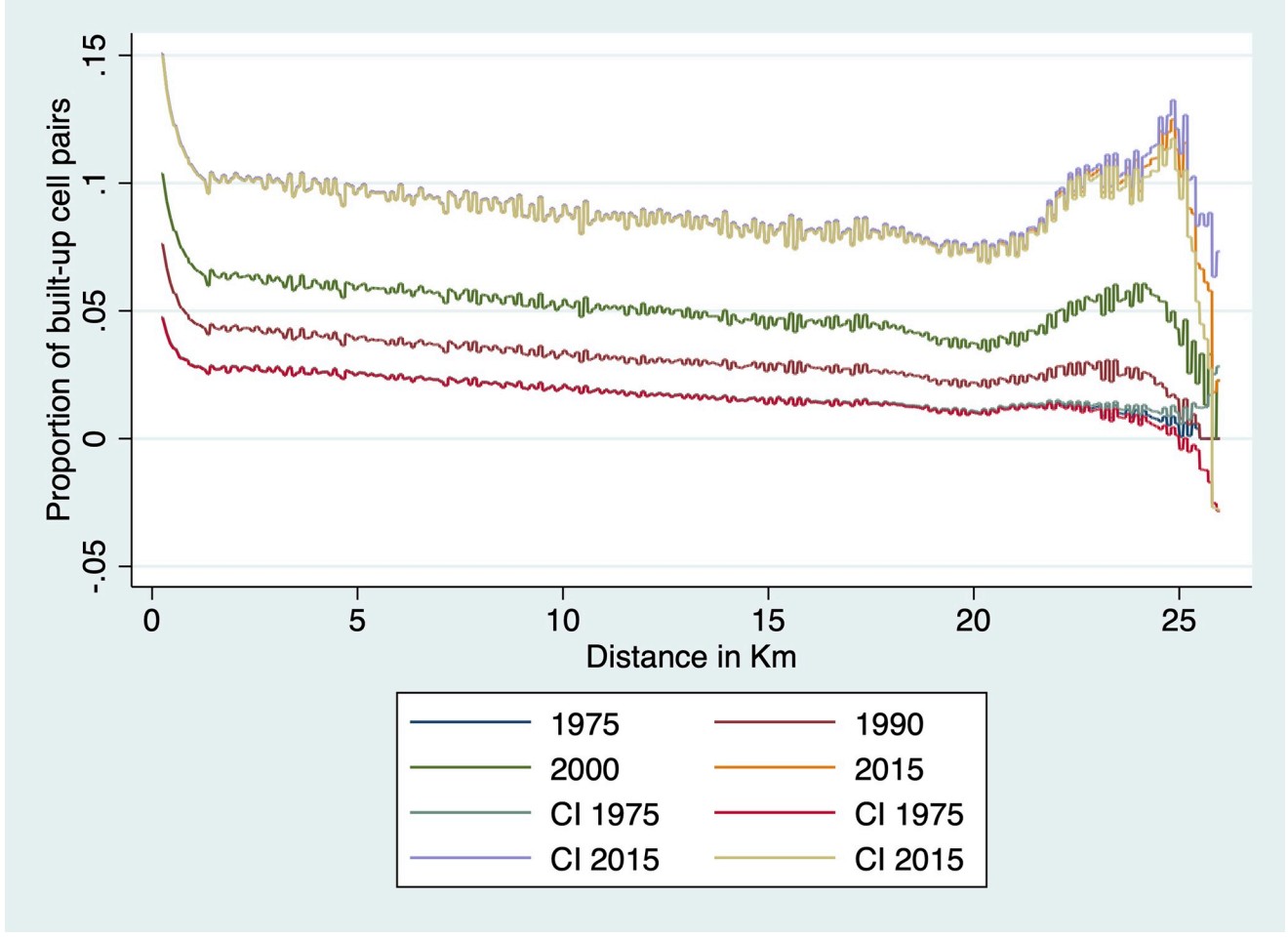

**Fig 12. Fig 12 is constructed in the same way as Fig 11, except that Fig 12 only includes cell pairs in which both cells are near a road, either in the cell itself or in a neighboring cell $n_1$ or $n_2$.** In Fig 12 we also show the 95% confidence interval for two of the years. The bounds of the confidence interval are only visible at long distances; for other values they are too tight to be visible on the graph.

any of these conjectures—or any equal spacing of villages up to 20 km. We do however find evidence of agglomerations covering an area of up to 3 km in radius—6 km or 4 miles in diameter. These localities may be larger than simple villages, and they may be located further apart than 20 km. Could there be, therefore, spikes in the spatial covariogram at distances above 20km—something that is possibly suggested by the appearance of a spike at distances slightly above 20km in Fig 12?

To investigate this possibility, we recalculate the spacial covariogram for distances up to 75 km using the procedure described in section (2.3.3). Results comparable to Figs 11 and 12 are shown in Figs 13 and 14, respectively. We see that the reported patterns are similar to those reported earlier. The fall in buildup correlation that was observed at short distances in Fig 11 is confirmed in Fig 13, and it is seem to continue then flatten our around about 50 km of distance. A similar pattern is shown in Fig 14 for cell pairs located near a road, except that the fall is steeper at short distances and the slope is steeper for longer. The curve also flattens our at around 50 km distance. We also observe that the bump at distances above 20 km in Fig 12 is not a robust result and is probably an artifact of the small number of tiles and cell pairs with observations at those distances. Using larger tiles solves that problem. These results confirm

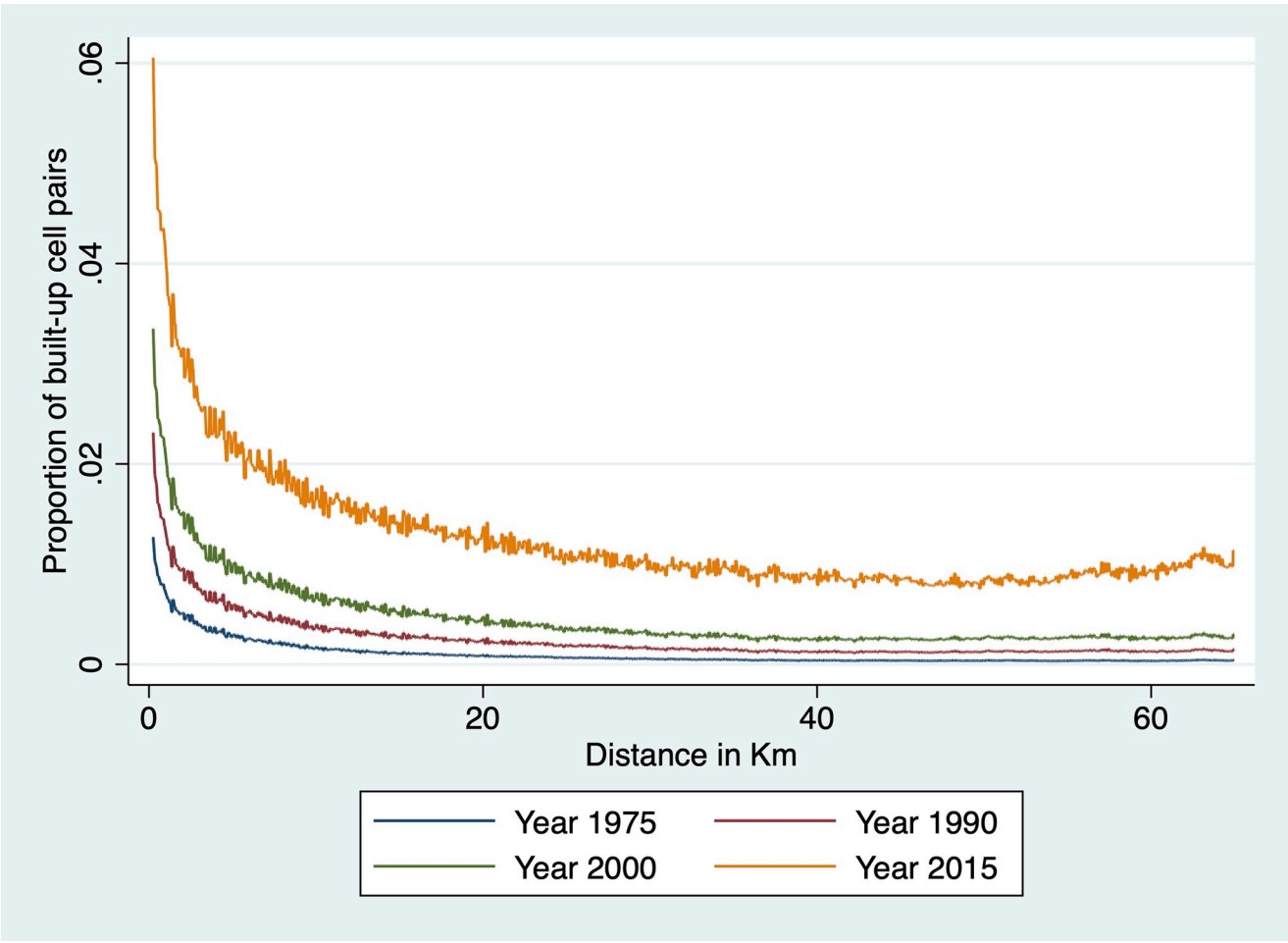

**Fig 13. Figs 13 and 14 show, for each distance value within big tiles, the proportion of cell pairs that are both built-up.** For instance, a value of 2% at distance 4Km means that pairs with two built-up cells make up 2% of all the cell pairs distant from each other by 4Km. Four correlograms are displayed, one for each of the years for which we have buildup data. The two Figures are constructed in the same way, except that Fig 14 only includes cell pairs in which both cells are near a road, either in the cell itself or in a neighboring cell $n_1$ or $n_2$. Confidence intervals are not shown because they are too tight to be visible in the graph.

our earlier findings, namely that, across Ghana, human settlements (as proxied by buildings) are not evenly spaced. This is true across cells in general, but also along roads. We do, however, find strong evidence of agglomeration at short distances, as well as evidence of weak agglomeration forces operating up 50 km away—in the sense that being less than 50 km from a built-up location predict a slightly higher probability of buildup. The figures also confirm that buildings tend to concentrate along roads and near each other.

In the last figure, we perform a similar exercise using small tiles as unit of observation. Since nearly all these tiles include at least one building, here we define buildup as being in the top decile in term of buildup, averaged over the whole tile. We then form all tile pairs and conduct the same analysis as in Figs 11 and 12. Distance between tiles is calculated from the tile's centroid which, for our purpose, is defined as the point with the average longitude and latitude of the tile. As we have much fewer observations, we need not put tile pairs into distance bins: we can work directly on the full sample of pairs. Given that nearly all tiles contain at least one road, we ignore that dimension here. Fig 15 shows the result with distance on the horizontal axis and, on the vertical axis, the share of tile pairs that are both in the top buildup decile. By

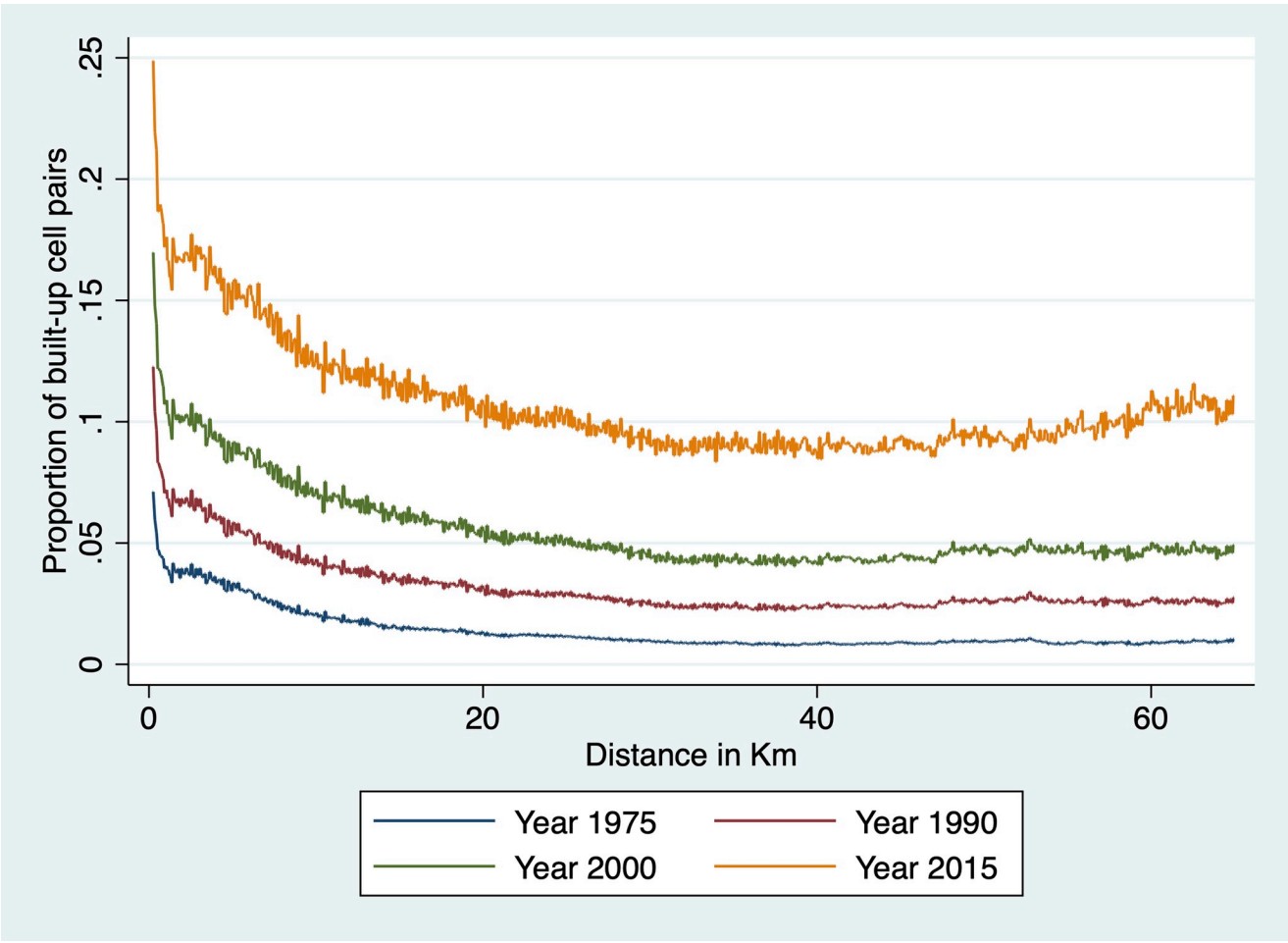

**Fig 14. Figs 13 and 14 show, for each distance value within big tiles, the proportion of cell pairs that are both built-up.** For instance, a value of 2% at distance 4Km means that pairs with two built-up cells make up 2% of all the cell pairs distant from each other by 4Km. Four correlograms are displayed, one for each of the years for which we have buildup data. The two Figures are constructed in the same way, except that Fig 14 only includes cell pairs in which both cells are near a road, either in the cell itself or in a neighboring cell $n_1$ or $n_2$. Confidence intervals are not shown because they are too tight to be visible in the graph.

construction, the horizontal axis is truncated at distances less than 18 km, which is the size of a tile itself. The four curves show, for each of the data year, the predictions from a fractional polynomial regression of the share of top decile pairs on distance. If buildup were distributed randomly across space, we should observe a flat curve. This is not what we see: as in Figs 11 to 14 we find evidence of agglomeration, which is strongest at distances up to around 60 km, which is the range of distances covered in Figs 13 and 14. Correlation in high buildup falls after that, but only gradually—suggesting that agglomeration continues to have an effect on buildup even at long distances.

## Conclusion

Developing countries across the world experienced strong population growth during the last four decades. With population growth, urbanization also picked up pace with half of the developing world's population now living in urban areas. The accelerated urbanization has received considerable attention in the Africa region where urbanization has not been associated as

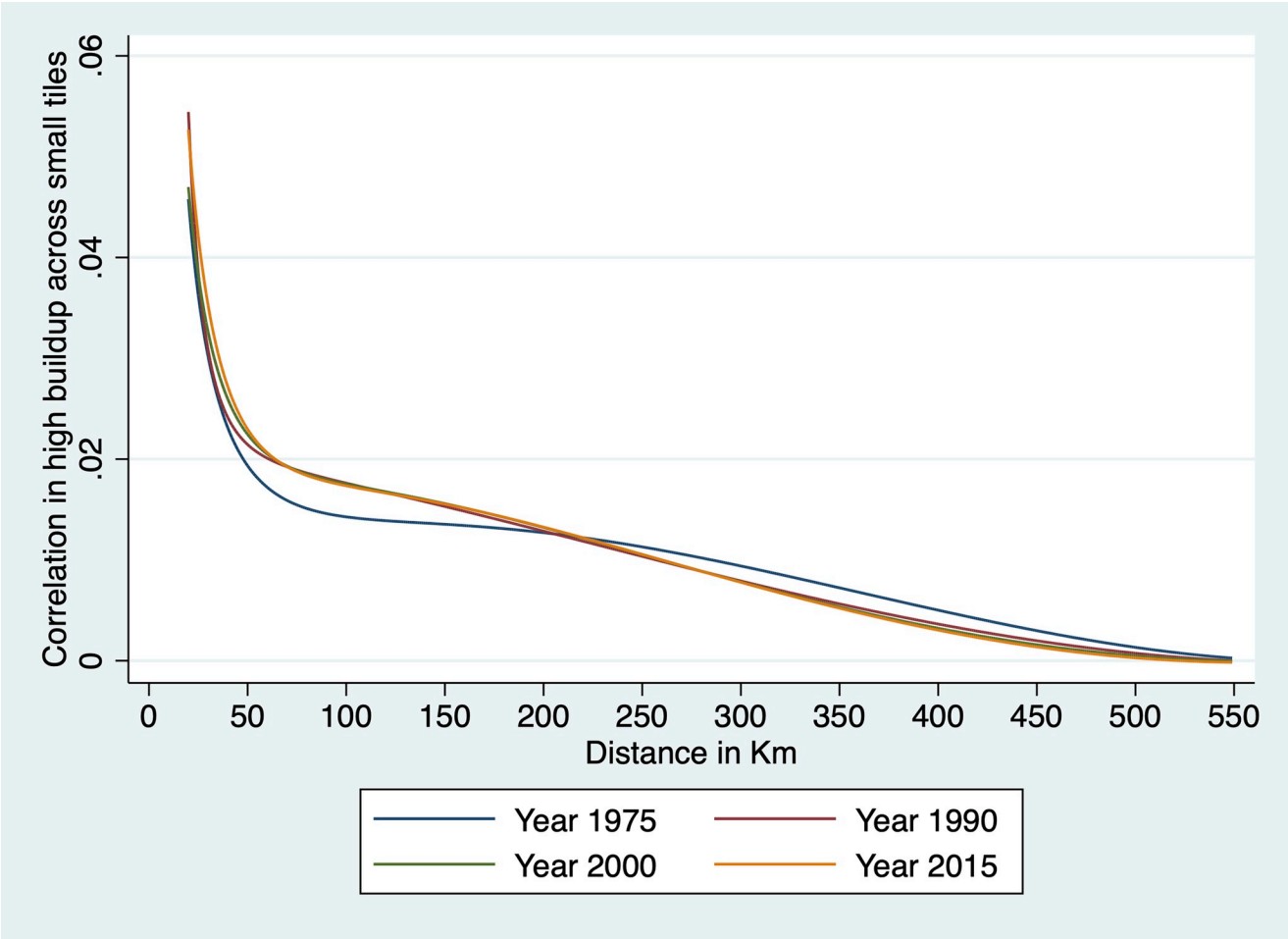

**Fig 15. Fig 15 shows, for each distance value between small tiles, the proportion of tile pairs that are both in the top buildup decile.** For instance, a value of 2% at distance 50Km means that tile pairs that are both in the top build decile make up 2% of all the cell pairs distant from each other by 50Km. Four correlograms are displayed, one for each of the years for which we have buildup data. The four curves show, for each data year, the predictions from a fractional polynomial regression of the share of top decile pairs on distance between pairs.

strongly with economic growth as observed historically in developed countries. For population growth to generate productivity and welfare gains in urban and rural areas, regional and city planners need to know how increased population will be distributed over geographical space as this is essential for planning provisions of infrastructure, housing and other public goods. The weaker link between urbanization and growth in Africa could be due to lack of provision of these services leading to haphazard urban growth, costly housing and low economic density. In this paper, we use high resolution satellite data on built-up areas within micro-pixels to study the evolution of human settlements in Ghana along the entire geographic continuum from major cities to smaller urban centers to rural areas. We study this from three distinct angles: forcasting of distribution of buildup areas overtime on the basis of observed pattern during last 40 years; within the closer range of 1.5 km focusing on agglomeration and persistence in buildup areas; and over a larger geographical area aiming to uncover regularity in the location of cities/towns in relation to each other.

Our analysis find that the proportion of fully built-up cells increased between 1975 and 2014 but the proportion of partially built-up cells remained essentially unchanged. This

suggests that, over the study period, cities grew at least partially at the expense of their hinterland We find strong evidence that agglomeration forces were at work, both within periods and across time as buildup in a cell predicts future buildup in itself and its surrounding cells. Roads proved to be a strong magnet for new settlements, with strong and lasting predictive effects associated with roads present at the beginning of our study period. Forecasts based on current agglomeration trends indicate that Ghana will experience considerable urban growth, and that this growth will take place mostly in concentrated fashion. We also expect a continuation of the current trend towards the rise of small and middle-size towns.

These findings have important implications for targeting provision of services in the coming years. The strong inertia in human settlement means that expansion in the provision of services can focus on the existing concentration of population, taking into account the fact that towns of all sizes, not just the large cities of Accra and Kumasi, will experience growth. Outside of towns and cities, the relatively low density of cultivation means that rural settlements are not regularly spaced, making it more difficult to plan the provision of rural amenities. But the fact that rural settlements increases fastest along roads means that expansion of provision will, at least initially, focus on roads. Since this is likely to further reinforce the tendency for people to locate near roads, opening new roads to currently unserved rural communities will be necessary to prevent a lopsided rural development pattern and ensure efficient usage of available agricultural land.

## Supporting information

**S1 Appendix.**
(DOCX)

## Acknowledgments

We thank for their useful comments the editor and four anonymous referees, as well as David Albouy, Mathilde Lebrand and participants to the 2020 Urban Economics Associate Virtual Meeting. We would like to acknowledge Varnitha Kurli for excellent research assistance in extracting all relevant GIS data used in the study. Our sincere thanks to Brian Blankespoor for his help with the GIS data. All views expressed in the paper are those of authors and should not be attributed to the Bank or its affiliates.

## Author Contributions

**Conceptualization:** Marcel Fafchamps.

**Data curation:** Forhad Shilpi.

**Formal analysis:** Marcel Fafchamps.

**Funding acquisition:** Forhad Shilpi.

**Investigation:** Marcel Fafchamps, Forhad Shilpi.

**Methodology:** Marcel Fafchamps.

**Project administration:** Forhad Shilpi.

**Software:** Forhad Shilpi.

**Visualization:** Marcel Fafchamps.

**Writing – original draft:** Marcel Fafchamps.

**Writing – review & editing:** Marcel Fafchamps, Forhad Shilpi.

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
