## [Decision Letter · Decision Letter 0]

20 Nov 2020

PONE-D-20-30587

The Evolution of Built-up Areas in Ghana since 1975

PLOS ONE

Dear Marcel,

Thank you for submitting your manuscript titled, ‘The Evolution of Built-up Areas in Ghana since 1975’ (Ref. No.: PONE-D-20-30587) to PLOS ONE. The manuscript has been reviewed. After careful consideration, we find that the manuscript has some merit with some interesting results, but it needs a major revision to address reviewers’ comments and fully meet PLOS ONE publication criteria. You can find reviewers’ comments at the bottom of this letter.

We invite you to submit a revised version of the manuscript. The changes required in the manuscript are very significant and require you to respond fully. We will send your revised manuscript for further external review. Therefore, we strongly recommend addressing concerns raised in full.

Be sure to address:

1. Restructure the manuscript, including a more convincing problem statement. One of the reviewers raised concern that the interpretation and contextualization of the work in housing policy and housing policy analysis in Africa seems weak. While the introduction reads well, it does not provide a gap in scientific knowledge that currently exists and why that gap needs to be filling?

2. Studies support using high-resolution imagery to study urban sprawl, in particular developing countries. Therefore, provide a convincing argument on using a coarse resolution data product that might have overestimated urban sprawl or even missed capturing a small extent of development. Repeat this study with high-resolution imagery for a few time steps and relate them with the corresponding coarse imagery products to show their accuracies.

3. Provide a “materials and methods” section with detailed descriptions of the study area, and steps used to accomplish the analysis. A majority of contents in results belong to the materials and methods section.

4. Currently, the manuscript draft reads like a report. Restructure various sections with substantive information to flow like a research article.

5. Provide a descriptive caption for each figure.

We would appreciate receiving your revised manuscript by January 19, 2020. When you are ready to submit your revision, log on to https://pone.editorialmanager.com/ and select the 'Submissions Needing Revision' folder to locate your manuscript file.

To enhance the reproducibility of your results, we recommend that if applicable you deposit your laboratory protocols in protocols.io, where a protocol can be assigned its own identifier (DOI) such that it can be cited independently in the future. For instructions see: http://journals.plos.org/plosone/s/submission-guidelines#loc-laboratory-protocols

We look forward to receiving your revised manuscript.

Regards,

Dr. Kunwar K. Singh

Academic Editor

PLOS ONE

2. Please include your tables as part of your main manuscript and remove the individual files. Please note that supplementary tables (should remain/ be uploaded) as separate "supporting information" files

3.We note that [Figure(s) A2.3, A2.2,  A2.1, A1.4, A1.3 and A1.1] in your submission contain map/satellite images which may be copyrighted. All PLOS content is published under the Creative Commons Attribution License (CC BY 4.0), which means that the manuscript, images, and Supporting Information files will be freely available online, and any third party is permitted to access, download, copy, distribute, and use these materials in any way, even commercially, with proper attribution. For these reasons, we cannot publish previously copyrighted maps or satellite images created using proprietary data, such as Google software (Google Maps, Street View, and Earth). For more information, see our copyright guidelines: http://journals.plos.org/plosone/s/licenses-and-copyright.

1.    You may seek permission from the original copyright holder of Figure(s) [A2.3, A2.2,  A2.1, A1.4, A1.3 and A1.1] to publish the content specifically under the CC BY 4.0 license. 

4.In your Data Availability statement, you have not specified where the minimal data set underlying the results described in your manuscript can be found. PLOS defines a study's minimal data set as the underlying data used to reach the conclusions drawn in the manuscript and any additional data required to replicate the reported study findings in their entirety. All PLOS journals require that the minimal data set be made fully available. For more information about our data policy, please see http://journals.plos.org/plosone/s/data-availability.

Reviewers' comments:

Reviewer's Responses to Questions

**Comments to the Author**

1. Is the manuscript technically sound, and do the data support the conclusions?

Reviewer #1: Yes

Reviewer #2: Partly

Reviewer #3: No

Reviewer #4: Yes

2. Has the statistical analysis been performed appropriately and rigorously? 

Reviewer #1: Yes

Reviewer #2: Yes

Reviewer #3: No

Reviewer #4: Yes

3. Have the authors made all data underlying the findings in their manuscript fully available?

Reviewer #1: Yes

Reviewer #2: Yes

Reviewer #3: No

Reviewer #4: Yes

4. Is the manuscript presented in an intelligible fashion and written in standard English?

Reviewer #1: Yes

Reviewer #2: No

Reviewer #3: No

Reviewer #4: Yes

5. Review Comments to the Author

Reviewer #1: Review: The Evolution of Built-up Areas in Ghana since 1975

The paper analyses the evolution of the built-up area in Ghana since 1975. The data and analysis make for good reading and are primarily sound. There are, however, three areas that I think are problematic (and the editor can decide how important this is for PLOS ONE).

• There is a minimal reference to similar work in Africa. I know of a paper published in “Nature” that did similar work for Africa.

• The paper is also not situated within the existing body of work. In addition, the paper could do more to situate the context within the housing research of Africa. Why is it important to study these aspects in Africa? The paper indeed references some of Henderson’s work, but there has been some recent work on the state of housing policy research in Africa.

• Linked to the above point, the interpretation and contextualisation (including the conceptual framework) of the work in housing policy and housing policy analysis in Africa seems weak. Often statements are made without much evidence and context – for example, the statement on urban sprawl in Ghana seems debatable when I look at the data. The question is, what are the 3-4 key issues of the paper – the existing overview is not good enough and haphazard to my mind.

• The section on Von Thunon’s as theoretical frame is not convincing. Surely times have changed – even in rural Africa. Alternatively, a more precise argument needs to be created around this issue

Reviewer #2: This study examined and described the Evolution of Built-up Areas in Ghana since 1975. The analysis carried out are very detailed and interesting whilst the results/findings in general are promising. I believe the outputs might be important for urbanists and other spatial planners. However, there is more room for improvement in many aspects of the “manuscript”. Therefore, before I can recommend it for acceptance and publication, it needs some major revisions.

May I advise that Authors pay attention to the following comments and suggestions including those provided in the manuscript (see attached pdf file).

1. I am afraid the submitted article simply does not read like a scientific paper. It reads more or less like a report of a sort. There is a distinctonly provided a detailed description of the data used.

2. The first part of the introduction was well written although the justification for this study was not very clear. My main concern however, is the latter part of the introduction which reads more or less like an “executive summary” of the study’s findings. Well, some of the key findings that have been presented in the results and discussion section were just repeated in the introduction and I simply do not understand why.

3. I am not too sure why the Von Thunen’s “agricultural land use” theory was chosen to study the “Evolution of Built-up Areas”. Maybe authors could explain and help us understand this better. As a geographer who has a broad knowledge of the Von Thunen’s theory, I do not think such a theory was appropriate for this study. I stand to be corrected though.

4. Also in the introduction, authors have indicated that the the paper contributes to the literature on urbanization and regional development citing Seto et al. 2011 and Henderson and Turner 2020 as reference literature. However, authors failed to elaborate on this. I was expecting some elaboration of this argument either in the discussion or even conclusion section. Authors should tell us what the literature on urbanization and regional development is saying; what and how is their present study contributing to it. As it is, the knowledge that is being added is not clearly spelt out.

5. Methodology wise, the article doesn’t provide any structure for readers to follow. In fact, there is no section devoted to materials and methods as stipulated in the authors guideline of PLOS ONE. Because of this, many of what can be described as methodological procedures have been presented in the results and discussion section. That is unacceptable in scientific work. May I suggest to the authors that they do their best to revisit the manuscript organisation page of PLOS ONE to see how their manuscript is expected to be structures: https://journals.plos.org/plosone/s/submission-guidelines#loc-parts-of-a-submission.

6. Although an elaborate results and discussion has been presented, it lacks understanding. Authors should clearly tell us what their results/findings imply or better still what the findings mean. As it it now, I simply cannot make any meaning out of what has been presented. All the methodological issues presented in this section should be moved to a “materials and methods” section.

7. The conclusion is too vague and lacks any substantive or analytical arguments in support of the key findings.

Reviewer #3: Hi Author

Thank you for the privilence to review this paper and below are some of my pointer why i think the paper is not ready for publication

1. The paper is poorly structured.

2. The paper is too long

3. There are not references in the paper (the athors were just giving an example of the paper), How did you cite your references?

4. 250 metre resoloution for urban studies is not accurate enough

5. Is the author is writing in first person or third person? There is a confusion in that regards

6. The reporting is not formal and its not good for academic publication.

7. The authors must work on the flow of the paper.

8. The different font types and font sizes needs to be looked into

9. The digrams are not clear and are not communciating with the main text.

10. if the lines on the paper were paper I was going to provide comments on each and every section with proper reference to the line number.

Hope this is clear.

Reviewer #4: This a is well-written paper that contributes significantly to the literature on urbanisation and urban transformation. The methodology is sound and replicable. The findings and conclusion are logically presented.

6. PLOS authors have the option to publish the peer review history of their article (what does this mean?). If published, this will include your full peer review and any attached files.

Reviewer #1: No

Reviewer #2: No

Reviewer #3: No

Reviewer #4: No

---

## [Author Response · Author response to Decision Letter 0]

17 Mar 2021

A full and detailed response to all the comments from the Editor and the four referees is included in the cover letter of our resubmission.

---

## [Decision Letter · Decision Letter 1]

15 Apr 2021

The Evolution of Built-up Areas in Ghana since 1975

PONE-D-20-30587R1

Dear Dr. Fafchamps:

We’re pleased to inform you that your manuscript has been judged scientifically suitable for publication and will be formally accepted for publication once it meets all outstanding technical requirements.

You’ll receive an e-mail detailing the required amendments shortly. When these have been addressed, you’ll receive a formal acceptance letter and your manuscript will be scheduled for publication.

An invoice for payment will follow shortly after the formal acceptance. To ensure an efficient process, please log into Editorial Manager at http://www.editorialmanager.com/pone/, click the 'Update My Information' link at the top of the page, and double check that your user information is up to date. If you have any billing related questions, please contact our Author Billing department directly at authorbilling@plos.org.

Best regards,

Dr. Kunwar K. Singh

Academic Editor

PLOS ONE

**Comments to the Author**

1. If the authors have adequately addressed your comments raised in a previous round of review and you feel that this manuscript is now acceptable for publication, you may indicate that here to bypass the “Comments to the Author” section, enter your conflict of interest statement in the “Confidential to Editor” section, and submit your "Accept" recommendation.

Reviewer #1: (No Response)

Reviewer #4: All comments have been addressed

2. Is the manuscript technically sound, and do the data support the conclusions?

Reviewer #1: Yes

Reviewer #4: (No Response)

3. Has the statistical analysis been performed appropriately and rigorously? 

Reviewer #1: Yes

Reviewer #4: (No Response)

4. Have the authors made all data underlying the findings in their manuscript fully available?

Reviewer #1: Yes

Reviewer #4: (No Response)

5. Is the manuscript presented in an intelligible fashion and written in standard English?

Reviewer #1: Yes

Reviewer #4: (No Response)

6. Review Comments to the Author

Reviewer #1: I think the authors have improved the paper. Yet, I still think the reflection on von Thunen does not add value

Reviewer #4: (No Response)

7. PLOS authors have the option to publish the peer review history of their article (what does this mean?). If published, this will include your full peer review and any attached files.

Reviewer #1: **Yes: **Lochner Marais

Reviewer #4: No

---

## [Editor Report · Acceptance letter]

11 May 2021

PONE-D-20-30587R1 

The evolution of built-up areas in Ghana since 1975 

Dear Dr. Fafchamps:

I'm pleased to inform you that your manuscript has been deemed suitable for publication in PLOS ONE. Congratulations! Your manuscript is now with our production department. 

Kind regards, 

on behalf of

Dr. Kunwar K. Singh 

Academic Editor

PLOS ONE